# Imprinted lncRNA Dio3os preprograms intergenerational brown fat development and obesity resistance

Yan-Ting Chen[1], Qi-Yuan Yang[1], Yun Hu[1], Xiang-Dong Liu[1], Jeanene M. de Avila [1], Mei-Jun Zhu[2], Peter W. Nathanielsz[3,4] & Min Du [1]✉

Maternal obesity (MO) predisposes offspring to obesity and metabolic disorders but little is known about the contribution of offspring brown adipose tissue (BAT). We find that MO impairs fetal BAT development, which persistently suppresses BAT thermogenesis and primes female offspring to metabolic dysfunction. In fetal BAT, MO enhances expression of Dio3, which encodes deiodinase 3 (D3) to catabolize triiodothyronine (T3), while a maternally imprinted long noncoding RNA, Dio3 antisense RNA (Dio3os), is inhibited, leading to intracellular T3 deficiency and suppression of BAT development. Gain and loss of function shows Dio3os reduces D3 content and enhances BAT thermogenesis, rendering female offspring resistant to high fat diet-induced obesity. Attributing to Dio3os inactivation, its promoter has higher DNA methylation in obese dam oocytes which persists in fetal and adult BAT, uncovering an oocyte origin of intergenerational obesity. Overall, our data uncover key features of Dio3os activation in BAT to prevent intergenerational obesity and metabolic dysfunctions.

[1] Nutrigenomics and Growth Biology Laboratory, Department of Animal Sciences, Washington State University, Pullman, WA 99164, USA. [2] School of Food Sciences, Washington State University, Pullman, WA 99164, USA. [3] Wyoming Pregnancy and Life Course Health Center, Department of Animal Science, University of Wyoming, Laramie, WY 82071, USA. [4] Southwest National Primate Research Center, Texas Biomedical Research Institute, San Antonio, TX 78227, USA. ✉email: min.du@wsu.edu

Obesity is prevalent in women at reproductive age[1]. In the United States, more than two-thirds of women of reproductive age are either obesity or overweight[2]. As a result, a large proportion of newborns in the United States and worldwide have experienced an intrauterine environment of mothers with obesity or overweight[3–5]. Epidemiological studies revealed that maternal obesity (MO) is associated with an increased risk of adolescent and adult obesity[6]. Alarmingly, obesity is linked to numerous chronic diseases, including type 2 diabetes, cardiovascular diseases, and several cancers[1,2,6]. Understanding underlying etiological mechanisms linking MO to offspring obesity is critical for curbing the current obesity epidemics.

Brown fat (BAT) is not only a key tissue for thermogenesis but also protects humans and mice from obesity and metabolic dysfunctions[7,8]. Compared with white adipose tissue (WAT), BAT has uncoupling protein-1 (UCP-1) anchoring in the inner mitochondrial membrane, dissipating the proton gradients for heat generation and increasing energy expenditure[7]. In humans and mice, impaired BAT thermogenesis leads to obesity, metabolic dysfunctions, and aging, whereas thermogenic activation enhances weight loss and insulin sensitivity[7,8]. The BAT thermogenesis is inducible, notably by cold stimulus, which depends on de novo formation of brown adipocytes from progenitor cells[9], which have fetal origins[10].

Lineage tracing studies demonstrate that BAT arises from Myf5+ myogenic cells during early embryonic development[11]. A set of master regulatory genes, including Prdm16 and Ebf2, commit these progenitor cells to the brown adipocyte lineage[11,12]. Genetic loss or blockage of Prdm16 or Ebf2 expression impedes fetal brown adipogenesis, which persistently impairs BAT thermogenesis in later life[11,12]. Thyroid hormones, including thyroxine (T4) and triiodothyronine (T3), are essential for fetal BAT development through activating thyroid hormone-responsive genes Ucp-1 and Ppargc-1a (protein PGC-1a)[13,14]. The absence of intracellular thyroid hormone signaling in fetal BAT permanently impairs BAT development and thermogenic function[15,16]. However, whether MO alters the thyroid hormone signaling during fetal BAT development remains unexamined. Interestingly, Dio3, encoding deiodinase 3 (D3), critical in regulating intracellular thyroid hormone homeostasis, is located in an imprinting locus[17,18].

Genomic imprinting is a phenomenon where a subset of genes demonstrate parental allele-specific expression[19]. The expression of imprinted genes is regulated by differential DNA methylation inherited from the parental germline[20]. Upon fertilization, despite of genome-wide DNA demethylation, the DNA methylation patterns of imprinting gene loci maintain during later development[21], suggesting that abnormal methylation marks in imprinting genes of germ cells may mediate intergenerational obesity. We hypothesized that MO alters the expression of imprinted genes regulating thyroid hormone metabolism, which affects brown adipogenesis during fetal development and generates "programming" effects on offspring BAT thermogenic functions later in life.

In the present study, we uncover that an imprinted long noncoding RNA (lncRNA) Dio3os inactivation increases D3 activity, resulting in T3 deficiency and impaired brown adipocyte differentiation via interfering PRDM16/PGC-1a complex. Enhancing Dio3os expression via adeno-associated virus (AAV) vector delivery in BAT activates brown adipogenesis and thermogenesis, protecting MO offspring from diet-induced adiposity and metabolic dysfunctions. We further find that the Dio3os promoter of obese dam oocytes has higher DNA methylation which persists in fetal and adult BAT, potentially explaining the suppression of Dio3os expression in MO fetal BAT. Our data identify a maternally imprinted lncRNA in attributing to intergenerational obesity through an oocyte-thermogenic fat axis, which provides therapeutic targets to prevent and treat inter/transgenerational obesity and metabolic dysfunctions due to MO.

## Results

**MO renders female offspring metabolic dysfunction and cold sensitivity.** After 10-wk of HFD feeding (45% calorie from fat), maternal mice developed MO as evidenced by >30% increase in body mass (Supplemental Fig. 1a), higher fasting blood glucose, insulinemia, and insulin resistance (Supplemental Fig. 1b–d). $O_2$ consumption and energy expenditure were also decreased in MO dam before mating (Supplemental Fig. 1e–h). Higher body mass was maintained to the end of lactation (Supplemental Fig. 1i).

After weaning, female and male offspring were fed normal or HFD for 12-wk at 22 °C (Fig. 1a). Regardless post-weaning diet, the female offspring were more susceptible to body weight gain and insulin resistance compared with male offspring due to MO (Supplementary Fig. 2a–f)[5,22]. MO male offspring also showed few alterations in body fatness, glucose sensitivity, $O_2$ consumption, and energy expenditure from normal or HFD feeding (Supplementary Fig. 2a–f). Notably, MO female offspring had similar caloric intake compared to CON offspring but gained more body mass (Fig. 1b, c), and also demonstrated glucose intolerance and insulin resistance compared with CON offspring (Fig. 1d, e). Metabolically, MO female offspring robustly displayed lower oxygen consumption and energy expenditure regardless of regression of body mass or lean mass (Fig. 1f, g and Supplemental Fig. 3a–d)[23]. In particular, the reduction of energy expenditure was worsened by accounting for body fat (Fig. 1g and Supplemental Fig. 3a, c and d), aligned with reduced core and surface body temperatures (Fig. 1h and Supplemental Fig. 3e, f), suggesting metabolic dysfunction of thermogenic fat. In consistency, MO female offspring had lower BAT mass (Fig. 1i) and contents of thermogenic protein UCP-1 and PGC-1a (Fig. 1j, k). Mitochondrial density (Fig. 1l and Supplemental Fig. 3g) and expression of genes involved in BAT thermogenesis and lipid oxidization were also decreased in MO offspring BAT (Fig. 1m), contributing to their excessive WAT deposition (Supplemental Fig. 3h, i). On the other hand, MO male offspring BAT mass and expression of thermogenic proteins were barely affected (Supplementary Fig. 2g, h), aligned with no change in body temperature (Supplementary Fig. 2i). These data show that MO negatively affects BAT functions primarily in female offspring, which were further examined.

To test BAT thermogenic potential, female offspring on a normal diet were subjected to 4 days of acute cold exposure at 4 °C (Fig. 2a). Food intake did not differ (Supplementary Fig. 4a) but MO offspring showed less gain of BAT mass and reduced inguinal-WAT browning under cold stimulus than control offspring (Fig. 2b and Supplementary Fig. 4b–e). Consistently, the expression of PRDM16 was suppressed in MO offspring BAT (Fig. 2c, d), so for UCP-1 and PGC-1a protein contents and expression of downstream adipogenic and thermogenic genes (Fig. 2c, d), explaining the reduced thermogenic capacity of MO female offspring (Fig. 2e, f).

To analyze the mediatory roles of BAT thermogenesis in energy metabolic dysfunction of MO female offspring, offspring were further acclimated to thermoneutrality (TN) at 30 °C to block BAT thermogenesis (Fig. 2a)[24]. At 30 °C, BAT thermogenic activity was inactivated as indicated by similar protein contents of UCP-1 and PGC-1a between control and MO female offspring (Fig. 2g and Supplementary Fig. 4f). TN promoted body weight gain and lipid accretion in WAT without significant difference between control and MO offspring (Fig. 2h–j). In addition, TN

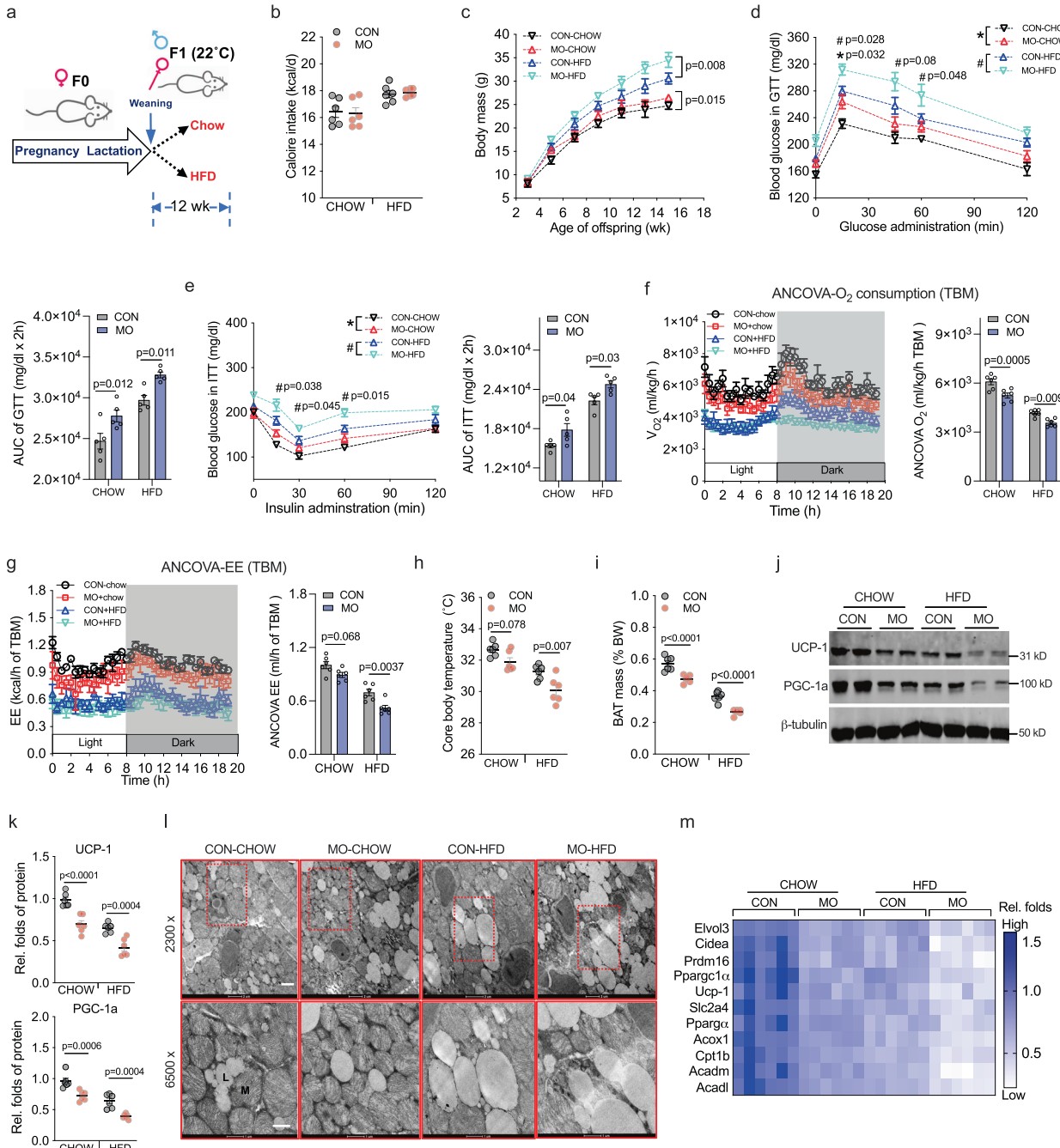

**Fig. 1 Maternal obesity (MO) reduces thermogenesis and energy expenditure of brown fat (BAT) and increases risks of metabolic dysfunctions in female offspring at 22 °C. a** At weaning, male and female offspring born to control and MO dam (C57BL/6 J) were fed a regular diet (chow, 10% calorie from fat) or high-fat diet (HFD, 60% calorie from fat) for 12 weeks (*n* = 6) at 22 °C. **b**, **c** Calorie intake (**b**) and body mass (**c**) of female offspring fed chow or HFD (*n* = 6). **d**, **e** After 6 h of fasting, female offspring were administrated with glucose or insulin for measuring glucose tolerance (GTT) (**d**) and insulin sensitivity (ITT) (**e**) (*n* = 5). * represents the difference between control and MO offspring fed chow diet, # represents the difference between control and MO offspring fed HFD, analysis was performed by Two-way ANOVA with adjustment-comparison test. **f**, **g** O$_2$ consumption (**f**) and energy expenditure (**g**) of female offspring (*n* = 6). Metabolic data were regressed to total body mass according to NIDDK MMPC ANCOVA analyses. **h** Measurement of core body temperature in female offspring rectum (*n* = 6). **i** Interscapular brown fat (BAT) mass (% body weight) in female offspring. **j**, **k** Abundance of UCP-1 and PGC-1a proteins in BAT measured by immunoblotting. β-tubulin was used as a loading control (*n* = 6). **l** Mitochondria and lipids in BAT were imaged by transmission electron microscopy (TEM) at ×2300 (scale bar: 2 μm) and ×6500 (scale bar: 1 μm) magnification (*n* = 4). **m** Heatmap shows mRNA expression of BAT thermogenic genes and genes in glucose and fatty acids uptakes. The mRNA expression was normalized to 18 S rRNA (*n* = 6). For data analyses, each pregnancy (dam) was considered as a replicate unit. Data are presented as mean ± s.e.m. Significance was determined by two-way ANOVA with Bonferroni post hoc analysis, and body mass curve was measured by two-way ANOVA with repeated measurements.

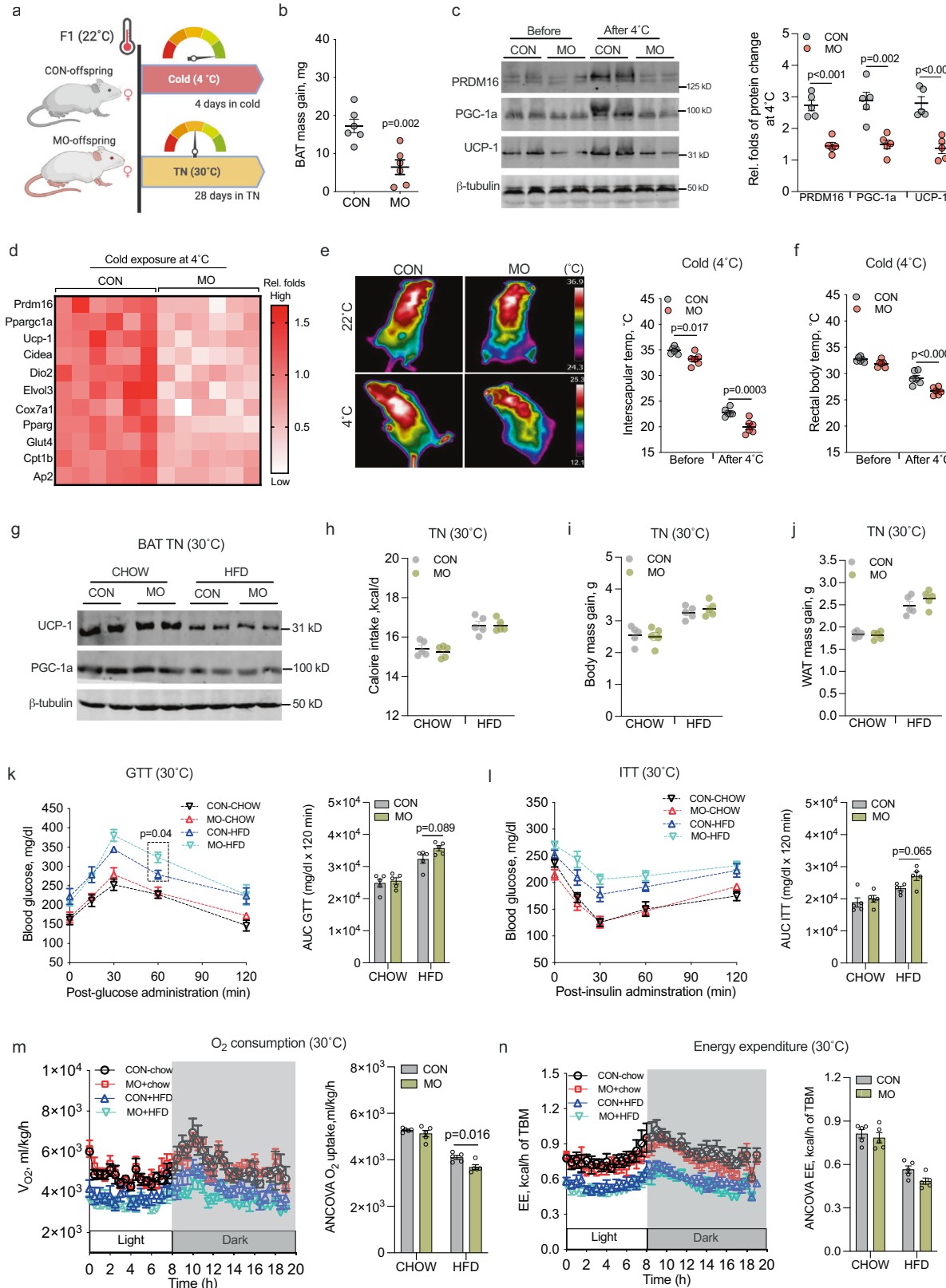

abolished the difference in glucose, insulin sensitivity, oxygen consumption, and energy expenditure between control and MO female offspring (Fig. 2k–n and Supplemental Fig. 4g, h). Taken together, these data showed that impairment of BAT thermogenesis mainly accounts for reduced energy expenditure and

metabolic dysfunction in MO female offspring under HFD and cold exposure.

Impaired brown adipogenesis in MO female offspring has a fetal origin. We next examined the developmental origin of BAT thermogenic impairment in MO female offspring. Reduced BAT

**Fig. 2 Maternal obesity (MO) impairs brown adipogenesis at 4 °C, that brown thermogenic inactivation and energy metabolic dysfunctions are ameliorated in female offspring at 30 °C. a** 4-month-old female offspring born from normal and MO dam was exposed to cold at 4 °C for 4 days ($n = 6$, chow diet) or acclimated to thermoneutrality (TN) at 30 °C for 28 days ($n = 5$, both chow and high-fat diet), respectively. **b** Offspring BAT mass gain after 4 °C exposure ($n = 6$). **c** Immunoblotting of adipogenic and thermogenic proteins in offspring BAT, including PRDM16, PGC-1a, and UCP-1. β-tubulin was used as a loading control ($n = 5$). **d** Heatmap shows mRNA expression of adipogenic and thermogenic genes in offspring BAT at 4 °C. mRNA expression was normalized to 18 S rRNA ($n = 6$). **e** Interscapular body temperature of offspring before and after 4 °C exposure. After removing mice from cages, thermal images were immediately taken to prevent heat loss and ambient temperature effects ($n = 6$). Animal behavior and scanning distance were controlled during image capturing. **f** Core body temperature of offspring measured in rectum using precise electronic thermometer ($n = 6$). **g** Thermogenic proteins UCP-1 and PGC-1a contents in BAT measured by immunoblotting after female offspring were acclimated to 30 °C for 28 days. β-tubulin was used as a loading control ($n = 5$). **h–j** Calorie intake (**h**), body mass gain (**i**), and white fat mass (**j**) of offspring at 30 °C ($n = 5$). **k, l** Offspring were intraperitoneally administered with glucose and insulin to test glucose tolerance (GTT) and insulin sensitivity (ITT) ($n = 5$). **m, n** Oxygen consumption (**m**) and energy expenditure (**n**) of offspring at 30 °C. Metabolic data were regressed to total body mass (TBM) according to guidelines of ANCOVA NIDDK MCCP tools ($n = 5$). Data are presented as mean ± s.e.m, each pregnancy (dam) was used as an experimental unit. Two-way ANOVA with Bonferroni post hoc analysis, and unpaired Student's $t$ test with two-tailed distribution were used in data analyses.

mass was not only observed in adults, but also in MO female offspring at weaning, and traced backed to fetuses at P0 and E18 (Supplemental Fig. 5a). Despite MO caused "fetal macrosomia" (Fig. 3a)[25], the ratio of BAT mass to body mass was persistently lower in MO female fetuses (Supplemental Fig. 5b), showing severe impairment of fetal BAT development and its long-term effects on offspring BAT function. Brown adipocyte precursors are developed during the fetal stage, which is required for pre- and postnatal BAT development[7,9,26]. We isolated stromal vascular fractions (SVFs) in fetal BAT and sorted for brown precursors using well-characterized brown adipogenic progenitor marker PDGFRα+[27] and brown preadipocyte marker EBF2[26,28]. Populations of brown progenitors (Lin−/CD45−/PDGFRα+) and preadipocytes (Lin−/PDGFRα+/EBF2+) were profoundly reduced (~50–70% decrease) in MO fetal BAT (Fig. 3b, c). Furthermore, the reduction persisted in MO offspring at weaning and 4 months of age (Fig. 3b, c). In consistency, isolated SVFs in MO fetal BAT had reduced capacity to differentiate into mature brown adipocytes in vitro (Supplementary Fig. 5c, d), explaining the reduced BAT development and plasticity in offspring BAT under the cold stimulus.

MO dysregulates T3 availability in fetal BAT. Thyroid hormones, T3 and T4, have been reported to activate brown adipogenesis and thermogenesis[29,30]. To confirm, we administrated thyroid hormones, T3, T4, or a combination of T3 and T4 to wild-type mice and then exposed them at 4 °C. Mice treated with thyroid hormones had increased BAT mass without affecting food intake and body weight (Supplementary Fig. 6a–c). Thyroid hormone administration increased the expression of thyroid hormone receptor (*THR*) and brown adipogenic and thermogenic genes *Prdm16*, *Ppargc-1a,* and *Ucp-1* (Supplementary Fig. 6d), which were aligned with higher interscapular and core body temperatures following cold exposure (Supplementary Fig. 6e–g).

We further examined whether MO altered thyroid hormone signaling in female fetal and neonatal BAT. MO did not alter T4 concentration but attenuated T3 in fetal and neonatal BAT (Fig. 3d). THRs *Thrα* and *Thrβ* and responsive genes *Ucp-1* and *Ppargc-1a* were robustly downregulated in MO fetal and neonatal BAT (Fig. 3e–g). MO neonatal BAT had lower mitochondrial density and thermogenic protein contents (Fig. 3g, h and Supplementary Fig. 4e), contributing to lower surface temperature and highly elevated expression of cold stress genes in the dorsal surface skin of neonates (Fig. 3i and Supplementary Fig. 5f, g). In alignment with the reduced BAT mass and brown precursors (Fig. 3a–c), *Prdm16* expression was also downregulated in MO fetal and neonatal BAT, contributing to an impaired brown adipocyte lineage determination (Fig. 3g and Supplementary Fig. 5h)[11].

Intracellular T3 concentration is regulated by extracellular T4 uptake from blood circulation, local conversion of T4 to T3, and further catabolization of T3 to inactive T2 (Fig. 3j). Iodothyronine deiodinase 2 (D2), encoded by *Dio2*, converts prohormone T4 to active T3 while iodothyronine deiodinase 3 (D3), encoded by *Dio3*, inactive T3 to T2 (Fig. 3j)[31,32]. T4, T3, and TSH concentrations in serum were not significantly altered in neither female fetuses, neonates (Supplementary Fig. 5i), nor dams due to MO (Supplemental Fig. 5j). MO also did not affect expression of MCT8, a membrane transporter for uptake of extracellular T3 and T4[31,32], in fetal and neonatal BAT (Supplemental Fig. 5k). Intriguingly, MO fetal and neonatal BAT had similar D2 content but profoundly higher D3 (Fig. 3k, l), showing excessive D3 expression might result in local T3 deficiency.

*Dio3os* inactivation blocks brown adipogenic differentiation via inducing T3 deficiency. Paternally expressed gene *Dio3*, together with maternally expressed lncRNA *Dio3os* are located in the *Dlk1-Dio3* imprinted locus (Fig. 4a)[17,18,33]. *Dio3os*, located ~1.6 kb upstream of *Dio3*[19,33,34], was suppressed in BAT of MO female fetuses and neonates (Fig. 4b), while other imprinted genes in the locus, such as *Dlk1*, *Rtl1*, *Glt2*, and most miRNAs, were barely altered (Supplemental Fig. 7a). Derived from the same progenitors and shared closer metabolic features with BAT[11], skeletal muscle in MO female fetuses also had lower expression of *Dio3os*, higher *Dio3* activation, aligned with lower content of T3, although such changes were significantly recovered after birth (Supplemental Fig. 7b–e). Meanwhile, we also assessed the development of sympathetic neurons flowing in female offspring BAT/WAT. Neural density, marked by tyrosine hydroxylase (TH), was reduced in inguinal-WAT of MO female offspring, but not in BAT (Supplemental Fig. 7h–j). Co-immunostaining of Dio3 protein with neurons, labeled by cholera toxin B (CTB) or TH, further showed limited overlapping, indicating *Dio3* has low expression in BAT/WAT neurons (Supplemental Fig. 7k–o). Meanwhile, no noticeable difference of *Dio3* and *Dio3os* expression was seen in BAT of male fetuses (Supplementary Fig. 7p) nor in other female organs, including WAT, liver, and heart (Supplementary Fig. 7b–g).

To assess *Dio3os* roles in brown adipogenesis, we used CRISPR-cas9 gRNA to knock down *Dio3os* expression in pluripotent mouse embryonic fibroblasts (MEFs), which was followed by brown adipogenic induction. *Dio3os* knockdown (~60%) (Fig. 4c) activated *Dio3* expression (Fig. 4d, e), which reduced intracellular T3 availability and expression of THR genes *Thrα* and *Thrβ* (Fig. 4f, g). Meanwhile, during brown adipogenic commitment, *Dio3os* knockdown blocked *Prdm16* expression (Fig. 4h) and brown adipogenic differentiation (Fig. 4i and Supplemental Fig. 8a). In consistency, fewer mature brown

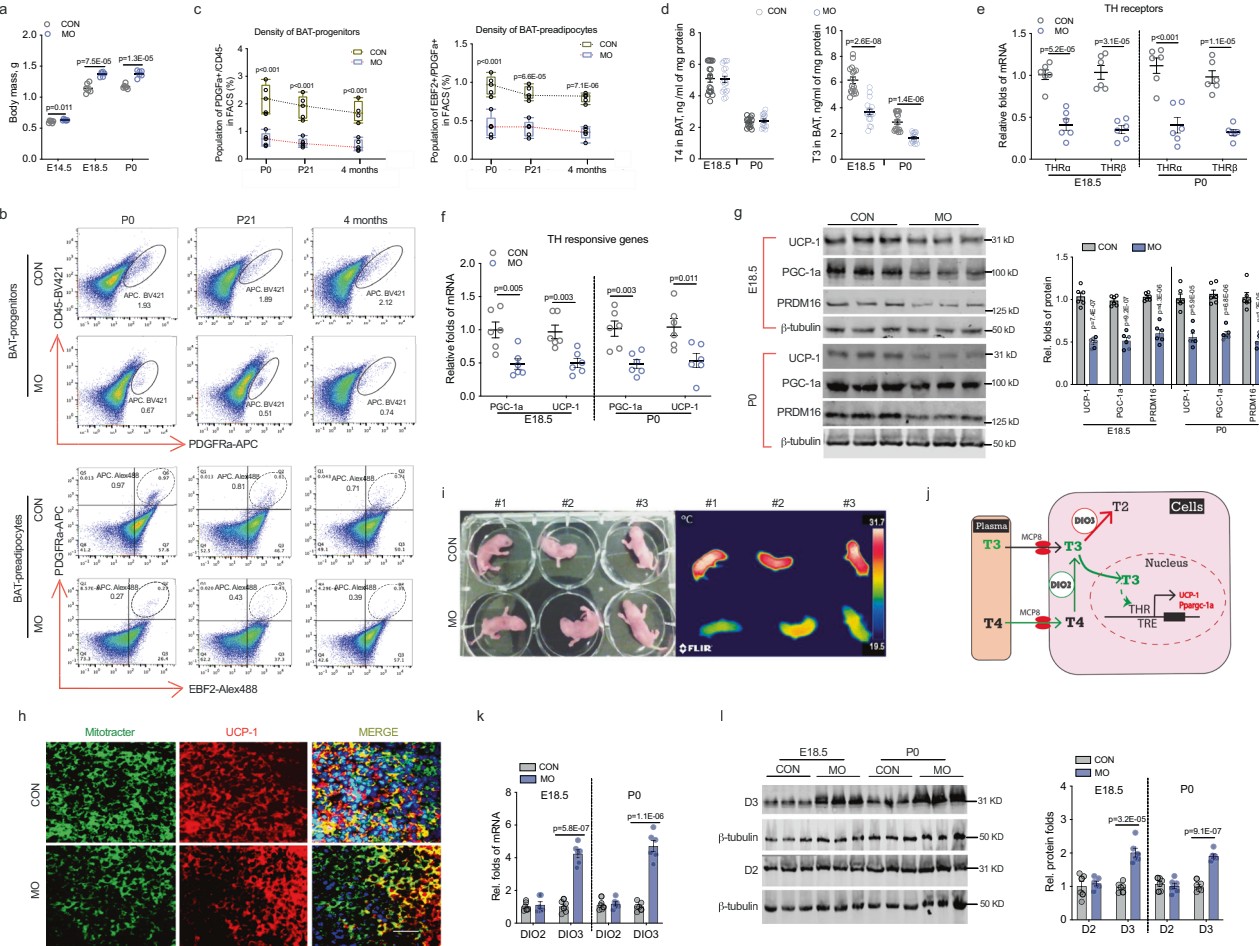

**Fig. 3 Maternal obesity (MO) reduces T3 concentration and thyroid hormone signaling, linking to impaired brown adipogenesis with a fetal origin. a** Fetal mass in control and obesity dam (MO) at 22 °C (n = 6). **b** Density of brown progenitors (solid circle) and preadipocytes (dash circle) in female offspring BAT at P0, 21 days (weaning), and 4 months of age. FACS sorted brown progenitors and preadipocytes in BAT-stromal vascular fractions using brown progenitor marker Lin: CD45−/PDGFRa+ (in black solid circle) and brown preadipocyte marker Lin: EBF2+/PDGFRa+ (dash black circle in right side quarter). **c** Quantified brown progenitors and preadipocytes in BAT of female offspring (n = 5). Whisker of box plots shows means and individual values from minimum to maximum. **d** Thyroid hormone (TH) thyroxine (T4) and triiodothyronine (T3) concentrations were analyzed in fetal and neonatal BAT at embryonic days E18.5 (n = 15) and P0 (n = 17). **e, f** mRNA expression of TH receptors TRα and TRβ (**e**), and TH responsive genes *Ppargc1a* and *Ucp-1* (**f**) in fetal and neonatal BAT at E18.5 and P0 (n = 6). mRNA expression was normalized to 18 S rRNA. **g** Immunoblotting measurements of PGC-1a, UCP-1, and PRDM16 protein contents in fetal (E18.5) and neonatal BAT (n = 6). β-tubulin was used as a loading control. **h** Immunostaining of mitochondrial (mito-tracker) and UCP-1 protein in neonatal BAT (n = 4). Scale bar 100 μm. **i** Thermal imaging of female neonates born to control and MO dam (n = 14 in control; n = 11 in MO). Imaging was taken immediately after removing from cages to limit the effects of ambient temperature and heat loss. **j** Diagram illustrates TH uptake, T3 and T4, from blood circulation and bio-active conversion in brown adipocytes. TH membrane receptor MCP8 for TH uptake in brown adipocytes, TH responsive element (TRE), nuclear TH receptors (THRα and β), iodothyronine deiodinases *Dio2* and *Dio3*, TH downstream thermogenic genes *Ucp-1*, *Ppargc1a*. **k** mRNA expression of *Dio2* and *Dio3* in fetal and female neonatal BAT at E18 and P0 (n = 6). mRNA expression was normalized to 18 S rRNA. **l** Protein contents of D2 and D3 were measured by immunoblotting; β-tubulin was used as a loading control (n = 6). For data analyses, each pregnancy (dam) was considered as a replicate unit. Data are presented as mean ± s.e.m. Unpaired Student's *t* test with two-tailed distribution was used in data analyses.

adipocytes were detected, and thermogenic protein expression and mitochondrial density were lower owing to *Dio3os* knockdown (Fig. 4j and Supplemental Fig. 8b–e). These changes were rescued by T3 addition (Fig. 4f–j and Supplemental Fig. 8b-e), showing mediatory roles of *Dio3os* suppression in limiting intracellular T3 level and brown adipogenesis.

PGC-1a protein competes with transcriptional suppressor CtBP1 to bind with PRDM16, which stably activates PRDM16 to promote brown adipocyte differentiation[35]. Because PGC-1a is a major thyroid hormone downstream effector, we further conducted co-immunoprecipitation of PRDM16 protein to assess whether *Dio3os* inactivation altered the interaction of PRDM16 with PGC-1a. *Dio3os*

knockdown increased protein CtBP1 but decreased PGC-1a binding to PRDM16 during brown adipogenic commitment (Fig. 4k). On the other hand, T3 supplementation robustly increased the binding of PGC-1a to PRDM16 at the expense of CtBP1 (Fig. 4k). Of significance, brown progenitors isolated in MO fetal BAT also displayed the reduced interaction of PGC-1a with PRDM16 but increased affinity of PRDM16 binding to CtBP1; such changes were also observed in adult BAT progenitors (Supplementary Fig. 8f, g and Fig. 4l), showing a persistent inhibition of PRDM16 brown adipogenic activity in MO female offspring BAT.

Because thyrotoxicosis can lead to embryonic lethality[36,37] and the placenta is impermeable to maternal thyroid hormones

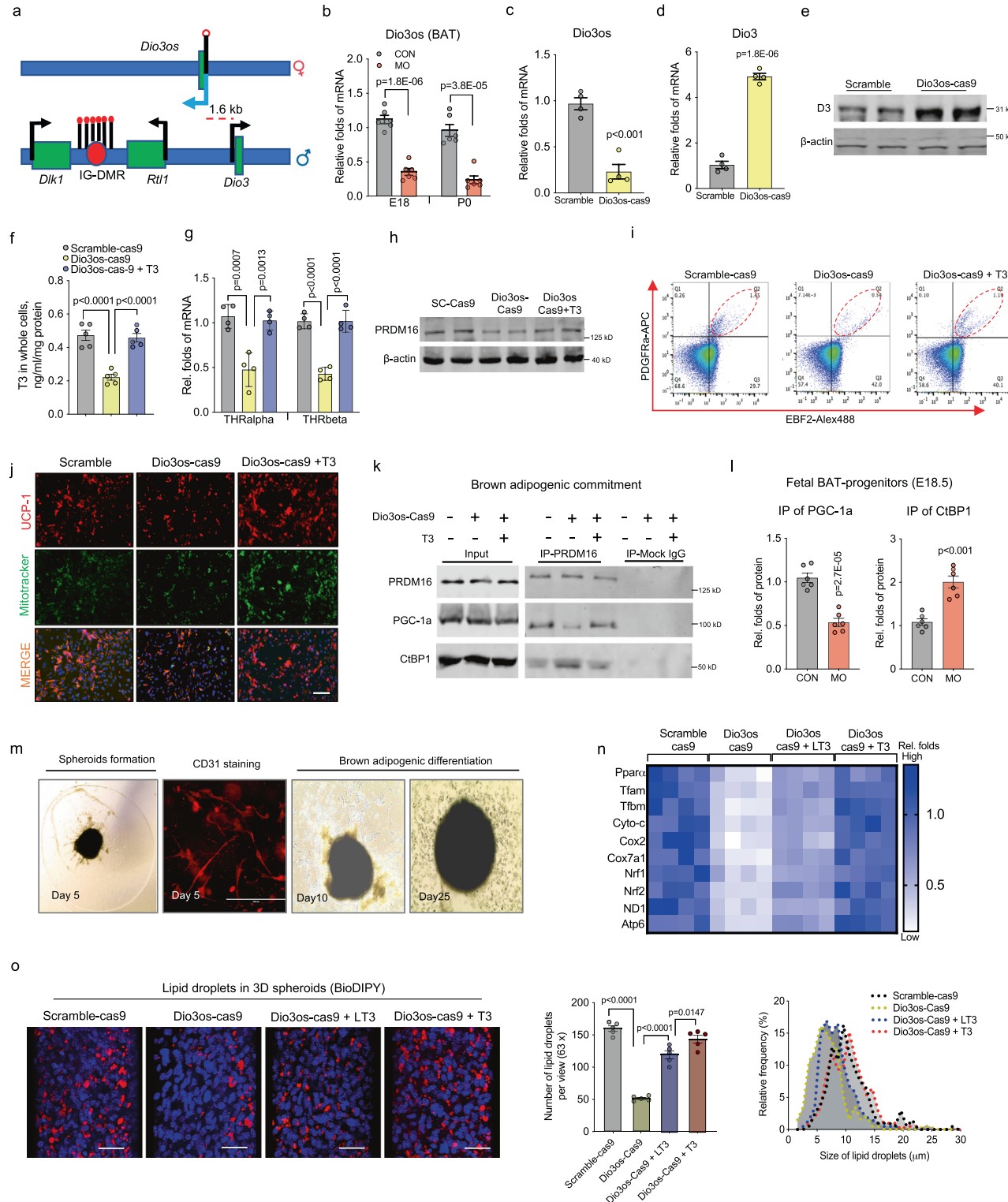

(0.008%)[38], we used a 3D ex vivo endothelial vascular network to validate the regulatory roles of *Dio3os* in brown adipocyte differentiation[39,40]. Mouse stromal vascular cells were isolated from fetal BAT and seeded into ultra-low attachment (ULA) dishes, and endothelial growth medium 2 (EGM2) was used to induce vasculature development in spheroids (Fig. 4m)[39,40]. After 5 days of culture, CD31-expressing endothelial cells assembled into a vascular network surrounding spheroids (Fig. 4m). *Dio3os* knockdown in spheroids promoted *Dio3* expression

(Supplementary Fig. 9a–c), which reduced intracellular T3 concentration (Supplementary Fig. 9d) and *Thrα* and *Thrβ* expression (Supplementary Fig. 9e). Such inhibition in spheroids was reversed by T3 supplementation (Supplementary Fig. 9d, e). T3 addition also recovered PRDM16 expression, and the formation of PGC-1a/PRDM16 protein complex (Supplementary Fig. 9f, g). After 20 days of brown adipogenic differentiation in Matrigel, *Dio3os* knockdown resulted in fewer lipid droplets in spheroids (Fig. 4o). Expression of thermogenic and oxidative

**Fig. 4 *Dio3os* suppression reduces T3 availability and PRDM16 activity, impairing brown adipogenesis in mouse embryonic fibroblasts and ex vivo organoids. a** Diagram shows the Dlk1-Dio3 imprinting locus. *Dio3os* is a maternally imprinted long-coding RNA (LncRNA), located ~1.6 kb upstream of the paternal *Dio3* gene. IG-DMR: imprinting control region with different DNA methylation. **b** mRNA expression of *Dio3os* in BAT of female fetuses and neonates at E18 and P0 (n = 6). Expression was normalized to 18 S rRNA. **c–n** Pluripotent mouse embryonic fibroblasts (MEFs) were transfected with Cas9 nuclease, CRISPR-tracrRNA, scramble, or Dio3os crRNA (Dio3os-cas9). MEFs were induced brown adipocyte differentiation for 6 days. MEFs in Dio3os-cas9 were also treated with 50 nM T3 (Dio3os-cas9 + T3) during brown adipocyte induction. **c** Following 72 h transfection, mRNA expression of *Dio3os* was measured in MEFs. Expression was normalized to 18 S rRNA (n = 4). **d**, **e** mRNA expression (**d**) and protein content of Dio3 (**e**) after 2 days of brown adipogenic commitment (n = 4). **f** T3 concentration in differentiated brown adipocytes (2 days) (n = 5). **g** mRNA expression of thyroid hormone receptors in differentiated brown adipocytes (2 days) (n = 4). **h** After brown adipogenic commitment for 2 days, immunoblotting measurement of PRDM16 protein content. β-actin was used as a loading control. **i** Population of committed brown preadipocytes (Lin: EBF2+/PDGFRa+) measured by flow cytometry sorting (FACS) (n = 4). **j** Immunostaining of mitochondria (mito-tracker) and UCP-1 in differentiated brown adipocytes (6 days) (n = 3). Scale bar 100 μm. **k** After brown adipogenic commitment for 2 days, PRDM16 was immunoprecipitated and protein lysates were further separated by SDS-PAGE. PRDM16, PGC-1a, and CtBP1 were measured by immunoblotting (n = 4). IgG was used as a negative control. **l** Quantification of PGC-1a and CtBP1 protein immunoprecipitated with PRDM16 in isolated brown preadipocytes of fetal BAT (E18.5) (n = 6). PRDM16, PGC-1a, and CtBP1 were detected by immunoblotting. IgG was used as a negative control. **m** Mouse BAT-stromal vascular fractions (SVFs) were induced to form vascular spheroids (5 days), and the presence of endothelial cells was visualized by CD31+ immunostaining. Spheroids were transfected with Cas9 nuclease, CRISPR-tracrRNA, scramble, or Dio3os crRNA (Dio3os-cas9) following brown adipocyte differentiation for 25 days. During brown adipogenic induction, Dio3os-cas9 spheroids were treated with low dose T3 (15 nM; Dio3os-cas9+ LT3) or high dose T3 (50 nM; Dio3os-cas9+ T3) (n = 4). **n** Heatmap displaying mRNA expression of mitochondrial biogenic and oxidative phosphorylation genes. mRNA expression was normalized to 18 S rRNA (n = 4). **o** Immunostaining staining of lipids using BioDIPY (red color) and nucleus using DIPY (blue color) (n = 4). Scale bar was 50 μm. Number and size of lipid droplets were quantified using image-J 3D project. Data are presented as mean ± s.e.m. Unpaired Student's t-test with two-tailed distribution and one-way ANOVA with adjusted multiple-comparison were used for data analyses.

phosphorylation genes was also reduced by *Dio3os* knockdown (Fig. 4n and Supplementary Fig. 9h). However, T3 supplementation dose-dependently rescued brown adipogenesis in spheroids with *Dio3os* knockdown (Fig. 4n, o and Supplemental Fig. 9f–h). Taken together, the data showed that *Dio3os* suppression blocks intracellular production of T3, impairing PRDM16 activity and brown adipogenesis.

To assess the causal effects, we further overexpressed either *Dio3os*, *Dio3*, or both in MEF, and then induced brown adipogenic differentiation (Supplementary Fig. 9i, j). *Dio3os* overexpression inhibited *Dio3* expression, which increased brown adipocyte formation and lipid accumulation (Supplemental Fig. 9j, k). Consistently, *Dio3os* overexpression activated brown adipogenic and thermogenic proteins, mitochondrial biogenesis, and oxygen consumption (Supplemental Fig. 9l–o). In contrary, *Dio3* overexpression blocked brown adipocyte differentiation and mitochondrial biogenesis (Supplemental Fig. 9k–o). Meanwhile, *Dio3os*-induced brown adipogenic activation was abrogated by *Dio3* overexpression (Supplementary Fig. 9k–o), showing *Dio3* suppression as the primary mechanism in activating brown adipocyte differentiation.

*Dio3os* activation in BAT increases PRDM16 activity and body energy expenditure in female mice. Based on in vitro and ex vivo findings, we further investigated *Dio3os* regulation in BAT thermogenesis and whole-body energy expenditure. We constructed *Dio3os* AAV (AAV-Dio3os), which was driven by a CMV promoter for *Dio3os* overexpression, and locally injected into interscapular BAT of female neonates to specifically activate *Dio3os* expression in BAT (Fig. 5a, d and Supplementary Fig. 10a)[41–43]. After 5-wk normal diet feeding at 22 °C, AAV-Dio3os mice had less body weight gain and increased BAT mass (Fig. 5b, c). *Dio3os* activation significantly reduced D3 content (Fig. 5d, f, g), leading to an increased T3 level in AAV-Dio3os-treated BAT (Fig. 5e). Meanwhile, AAV-Dio3os administration increased multi-lipid droplets in adipocytes (Fig. 5h), and activated BAT thermogenic genes and mitochondrial biogenesis (Fig. 5f–i and Supplementary Fig. 10b), and also increased affinity of PGC-1a protein binding to PRDM16 at the expense of CtBP1 in promoting brown adipogenesis (Fig. 5j). In response to BAT thermogenic activation, AAV-Dio3os mice showed an elevated body energy expenditure and oxygen respiration (Fig. 5k, l). AAV-Dio3os mice displayed

higher body temperature under cold exposure at 4 °C (Fig. 5m, n). Taken together, *Dio3os* activates brown adipogenesis and thermogenesis, which increases whole-body energy expenditure.

*Dio3os* activation in BAT prevents MO female offspring from high-fat diet (HFD)-induced obesity. To uncover the roles of *Dio3os* inactivation in mediating MO female offspring energy expenditure and obesity, we further injected AAV-Dio3os-CMV into MO female offspring interscapular BAT to recover *Dio3os* expression (Fig. 6a, b). After 5-wk HFD feeding at 22 °C, MO offspring with AAV-Dio3os administration showed resistance to body weight gain and increased BAT mass compared with MO offspring without AAV-Dio3os administration (Fig. 6c, d and Supplemental Fig. 10c). Meanwhile, MO offspring BAT with AAV-Dio3os displayed higher expression of adipogenic and thermogenic proteins (Fig. 6e, f and Supplemental Fig. 10d, f), correlated with less lipid deposition in subcutaneous and visceral WAT, and liver (Fig. 6e, g and Supplemental Fig. 10e). BAT *Dio3os* activation also substantially alleviated impaired insulin sensitivity, mitochondrial biogenesis, and respiration in MO offspring WAT and liver (Supplemental Fig. 10g-k), showing essential roles of BAT thermogenesis in maintaining WAT- and hepatic metabolic health. Furthermore, MO offspring with AAV-Dio3os had improved whole-body glucose tolerance and insulin sensitivity (Fig. 6h and Supplemental Fig. 10l, m), as well as increased oxygen consumption and energy expenditure (Fig. 6i and Supplemental Fig. 10n). To test cold resistance, MO female offspring with AAV-Dio3os were further exposed at 4 °C for 4 days. Under the cold stimulus, MO offspring with AAV-*Dio3os* recovered adipogenic capability to gain BAT mass compared with control offspring (Fig. 7a). AAV-Dio3os improved BAT thermogenic activity and mitochondrial biogenesis in MO female offspring (Fig. 7b and Supplementary Fig. 11a, b), aligned with improved glucose sensitivity and cold resistance (Fig. 7c–e).

In addition, we acclimated MO female offspring with AAV-Dio3os administration at 30 °C for 5-wk. Under TN, *Dio3os* activation in BAT blocked *Dio3* expression and increased T3 availability (Supplemental Fig. 11c–e). BAT mass was not altered, but AAV-Dio3os increased expression of BAT thermogenic proteins, which limited excessive body weight gain and lipid deposition in inguinal- and gonadal-WAT (Fig. 7f–h and Supplementary Fig. 11f–j). Consistently, AAV-Dio3os improved

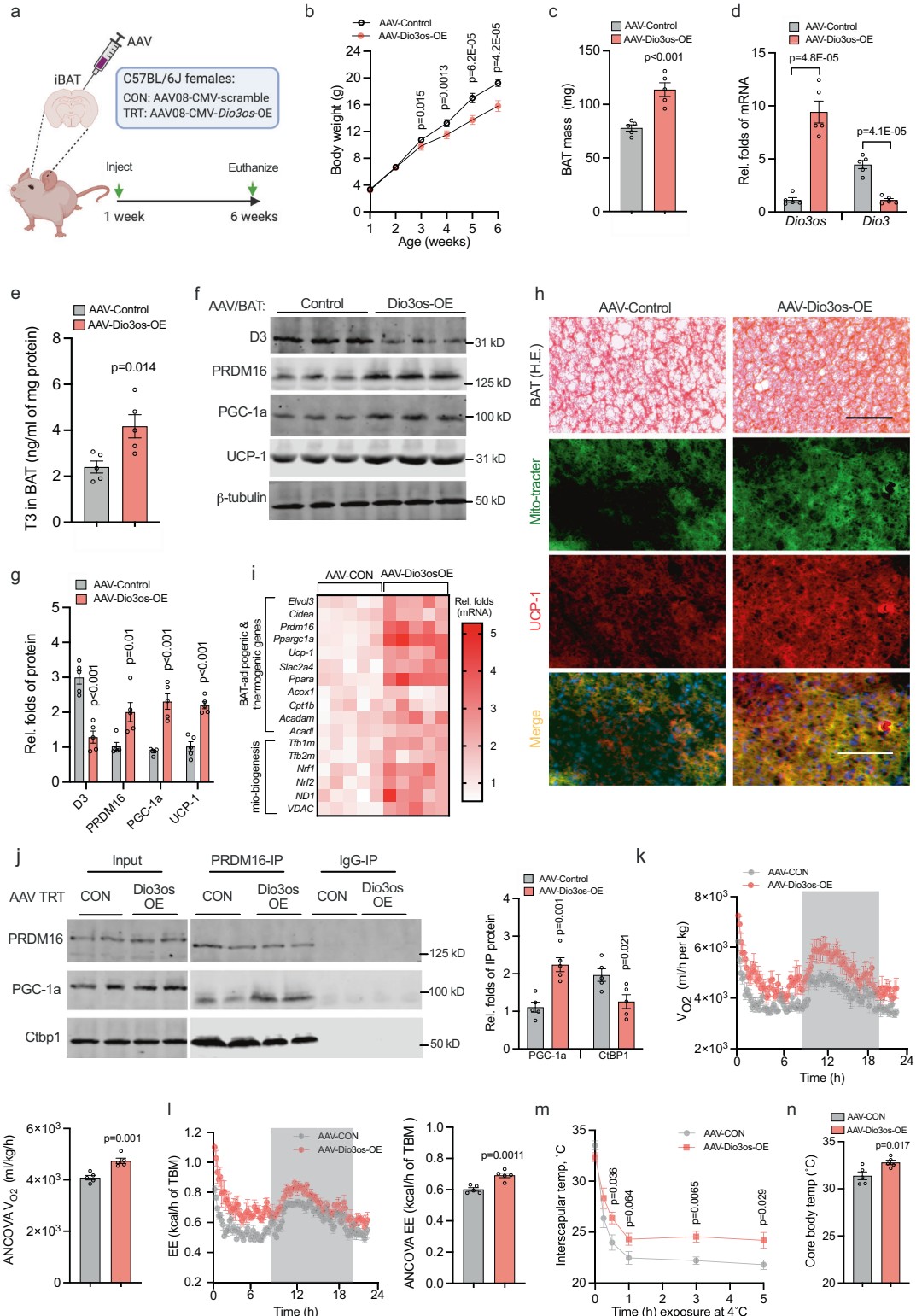

glucose sensitivity, $O_2$ consumption, and energy expenditure of MO offspring (Fig. 7i–k). Taken together, these data showed that *Dio3os* activation in BAT improves energy expenditure and glucose tolerance of MO female offspring in cold and TN.

*Dio3os* hypermethylation mediates *Dio3os* suppression in obese dam oocytes and female offspring BAT. DNA methylations in imprinted genes are protected from erasing during gamete fertilization and earlier embryogenesis[18,19]. The *Dio3os* promoter has

rich CpG sites, which are highly conserved between humans and mice (Fig. 8a). MO fetal BAT displayed dense 5'-methylcytosine (5mC) in the *Dio3os* promoter either assayed by MeDIP or bisulfite pyrosequencing (Fig. 8b, d and e), which was associated with transcriptional suppression (Fig. 8c). However, 5mC abundances in imprinting genes of the same cluster, *Dlk1*, *Rtl1*, and *Gtl2*, and IG-DMR were barely altered (Supplemental Fig. 12a–c). Intriguingly, higher methylation of the *Dio3os* promoter was also

**Fig. 5 In vivo *Dio3os* expression activates brown adipogenesis and thermogenesis, improving energy expenditure and glucose tolerance. a** C57BL/6 J female mice at 7 days of age (n = 5) were injected with adeno-associated virus type 8 (AAV8)-*Dio3os*-CMV or AAV8-scramble-CMV (1.5 × 10¹¹ vg per mouse) into interscapular brown fat (BAT) at 22 °C. Mice were euthanized 5 weeks post-injection at 22 °C. **b, c** Body weight (**b**) and BAT mass (**c**) after AAV injections (n = 5). **d** mRNA expression of *Dio3os* and *Dio3* in interscapular BAT at 5 weeks post-injection (n = 5). **e** T3 concentration (**e**) in BAT (n = 5). **f, g** Immunoblotting measuring protein contents of D3, PRDM16, PGC-1a, and UCP-1 in BAT (n = 5). β-tubulin was as a loading control. **h** H&E staining and immunohistochemical staining of mitochondria (mito-tracker, green) and UCP-1 (red) in BAT (n = 5). Scale bar 150 μm (H&E) and 250 μm (immunostaining). **i** Heatmap displaying mRNA expression of brown adipogenic and mitochondrial biogenic genes in BAT (n = 5). **j** PRDM16 immunoprecipitation in measuring PGC-1a and CtBP1 complex in isolated BAT-SVFs. PRDM16, PGC-1a, and CtBP1 were detected by immunoblotting. IgG was used as a negative control (n = 5). **k, l** Oxygen consumption (**k**) and energy expenditure (**l**) of mice at 5-week post AAV injections (n = 5). Metabolic data were regressed to body mass according to guidelines of ANCOVA NIDDK MCCP tool. **m, n** Mice were exposed to 4 °C and surface body temperature (**m**) and core body temperature (24 h in cold) (**n**) (n = 5). Data are presented as mean ± s.e.m. Unpaired Student t test with two-tailed distribution was used for data analyses.

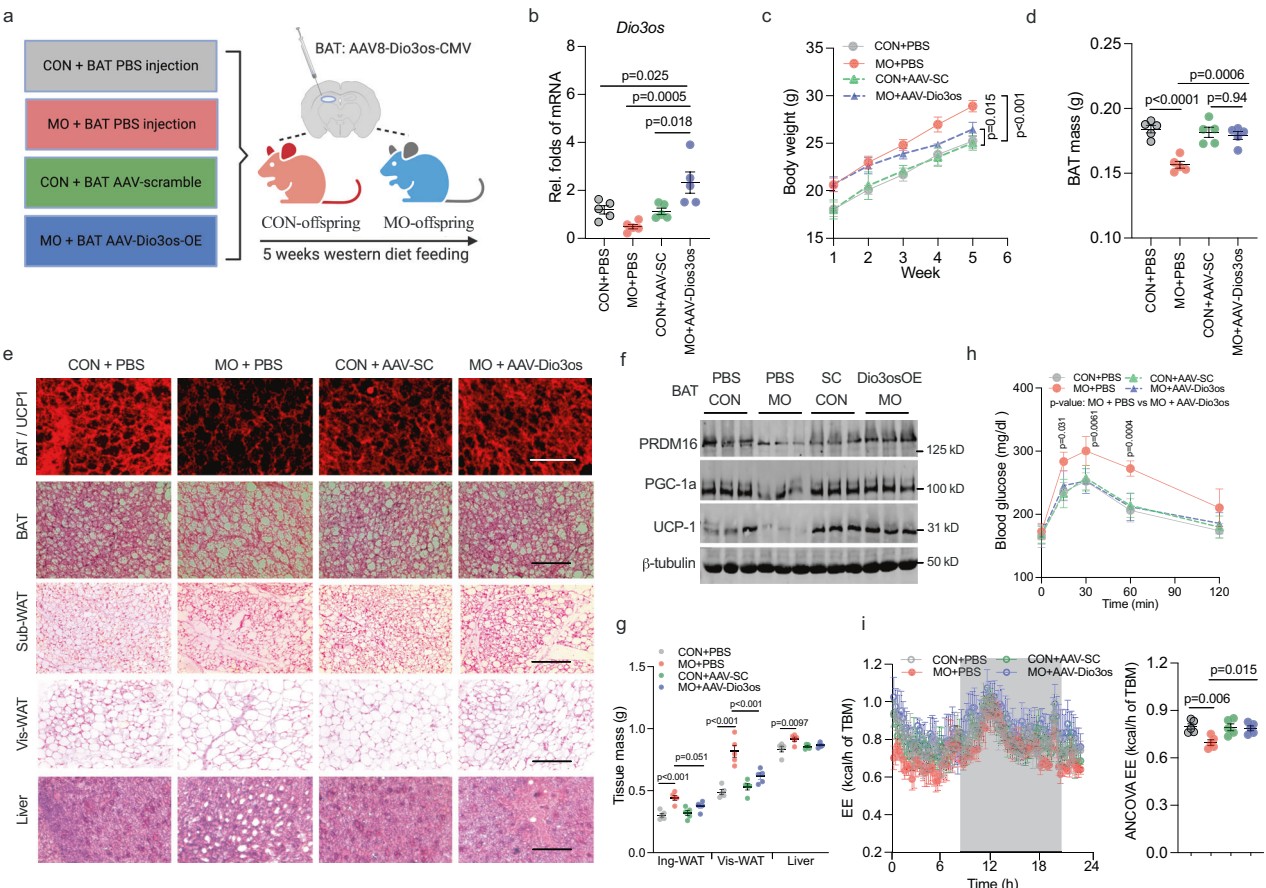

**Fig. 6 *Dio3os* activation in brown fat (BAT) protects maternal obesity (MO) female offspring from western-diet-induced obesity and fatty liver at 22 °C. a** 8-week-old female offspring born from normal and MO dams were injected with AAV8-scramble-CMV or AAV8-Dio3os-CMV viral particles (5 × 10¹¹ vg per mouse) into interscapular BAT, respectively. Control and MO offspring at the same age were treated by saline injection into interscapular BAT as negative controls. After injection, offspring were fed HFD for 5 weeks at 22 °C. **b** mRNA expression of *Dio3os* in offspring BAT at 5-week post injection (n = 5). **c, d** body weight (**c**) and BAT mass (**d**) in offspring post-injection (n = 5). **e** UCP-1 immunostaining in offspring BAT at 5-wk post injection. H&E staining of BAT, subcutaneous, visceral white fat, and liver. Scale bar 250 μm (immunostaining) and 200 μm (H&E) (n = 5). **f** Immunoblotting measuring protein contents of PRDM16, PGC-1a, and UCP-1 in offspring BAT at 5-wk post-injection (n = 5). β-tubulin was the loading control. **g** Inguinal, visceral fat and liver mass in offspring at 5-wk post injection (n = 5). **h** Blood glucose in offspring after intraperitoneal glucose injection for measuring glucose tolerance (n = 5). **i** Offspring energy expenditure at 5-week post injection (n = 5). Metabolic data were regressed to body mass according to guidelines of ANCOVA NIDDK MCCP tool. Data are presented as mean ± s.e.m. Two-way ANOVA with Bonferroni post hoc was used in data analysis.

observed in adult BAT (Fig. 8e–f and Supplemental Fig. 12d), contributing to the reduced *Dio3os* expression in MO adult offspring (Supplemental Fig. 12e). Consistently, BAT in MO adult offspring displayed higher *Dio3* expression, and reduced T3 concentration and expression of THRs (Supplemental Fig. 12f–h).

Meanwhile, the higher methylation of *Dio3os* promoter was also observed in obese dam oocytes before mating (Fig. 8g), which correlated with reduced *Dio3os* expression (Fig. 8h), higher *Dio3* expression, and D3 protein content (Fig. 8I, k). Obese dam oocytes consistently showed less mtDNA (Fig. 8j). In agreement

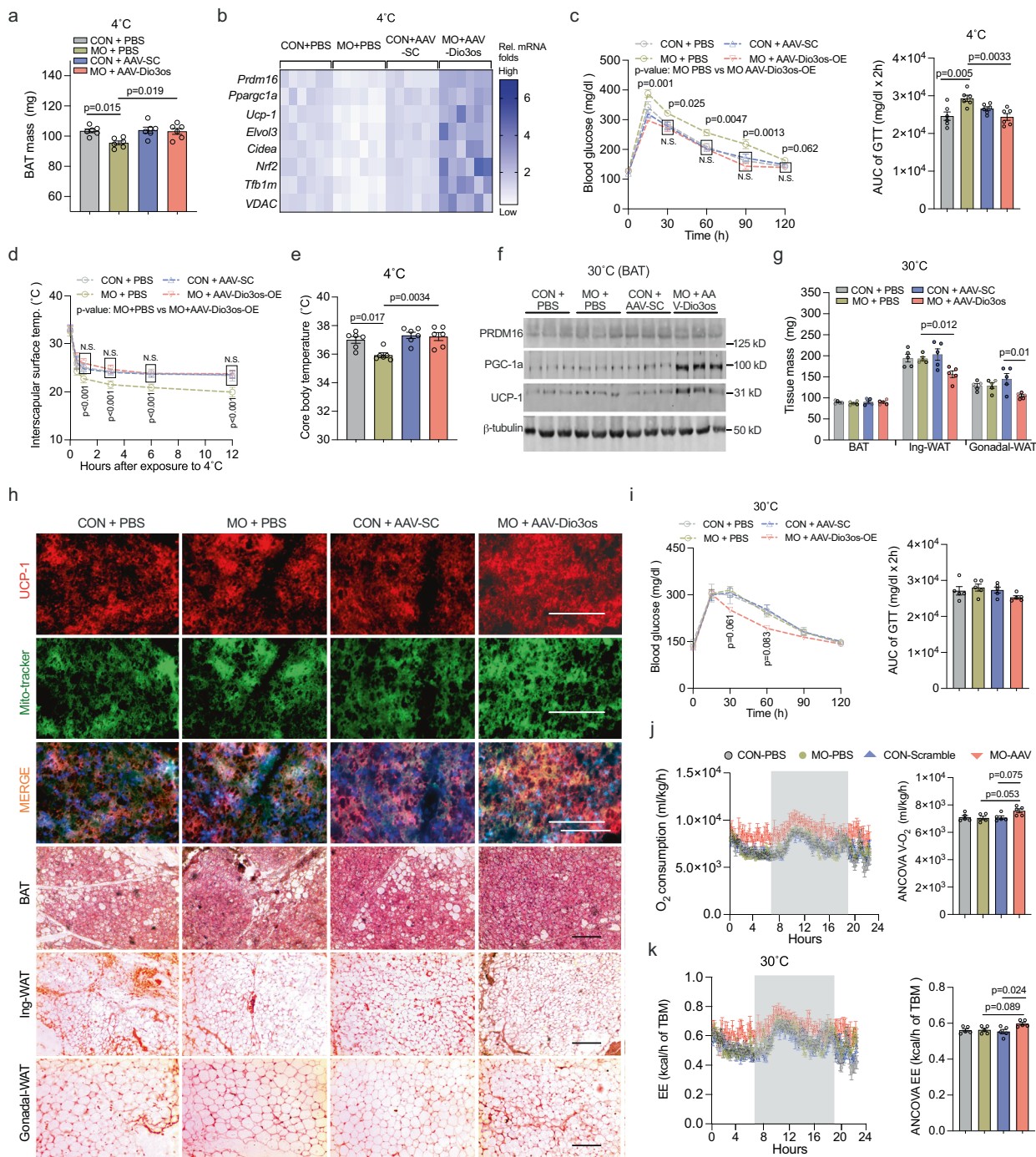

**Fig. 7 *Dio3os* activation in BAT improves MO female offspring 4 °C-cold resistance, and energy expenditure and glucose tolerance at 30 °C. a** After 4 °C exposure for 4 days, BAT mass in 8-week-old normal and MO female offspring administrated with PBS, AAV8-scramble-CMV or AAV8-Dio3os-CMV viral particle in BAT ($1.5 \times 10^{11}$ vg per mouse, $n = 6$). **b** Heatmap displaying mRNA expression of brown thermogenic and mitochondrial biogenic genes in BAT after 4 °C exposure ($n = 6$). mRNA expression was normalized to 18 S rRNA. **c** Glucose tolerance of offspring at 4 °C ($n = 6$). **d, e** Interscapular surface (**d**) and rectal body temperature (**e**) in offspring at 4 °C ($n = 6$). **f–j** Normal and MO female neonates were administrated with PBS, AAV-scramble, or AAV-Dio3os in BAT ($1.5 \times 10^{11}$ vg per mouse) and further acclimated to 30 °C for 5-wk ($n = 5$). **f** Immunoblotting measuring PRDM16, UCP-1, and PGC-1a proteins in offspring BAT at 30 °C ($n = 5$). β-tubulin was used as a loading control. **g** BAT, inguinal- and gonadal-WAT mass in offspring at 30 °C ($n = 5$). **h** Immunostaining of UCP-1 and mito-tracker in BAT, and H&E staining of BAT, inguinal- and gonadal-WAT ($n = 5$). Scale bar 250 µm in immunostaining and 200 µm in H&E. **i** Blood glucose concentration after intraperitoneal glucose administration in offspring for glucose tolerance test at 30 °C ($n = 5$). **j, k** O$_2$ consumption (**j**) and energy expenditure (**k**) in offspring at 30 °c ($n = 5$). Metabolic data were regressed to body mass according to guidelines of ANCOVA NIDDK MCCP tool. Data are presented as mean ± s.e.m. Two-way ANOVA with Bonferroni post hoc analysis was used in data analysis.

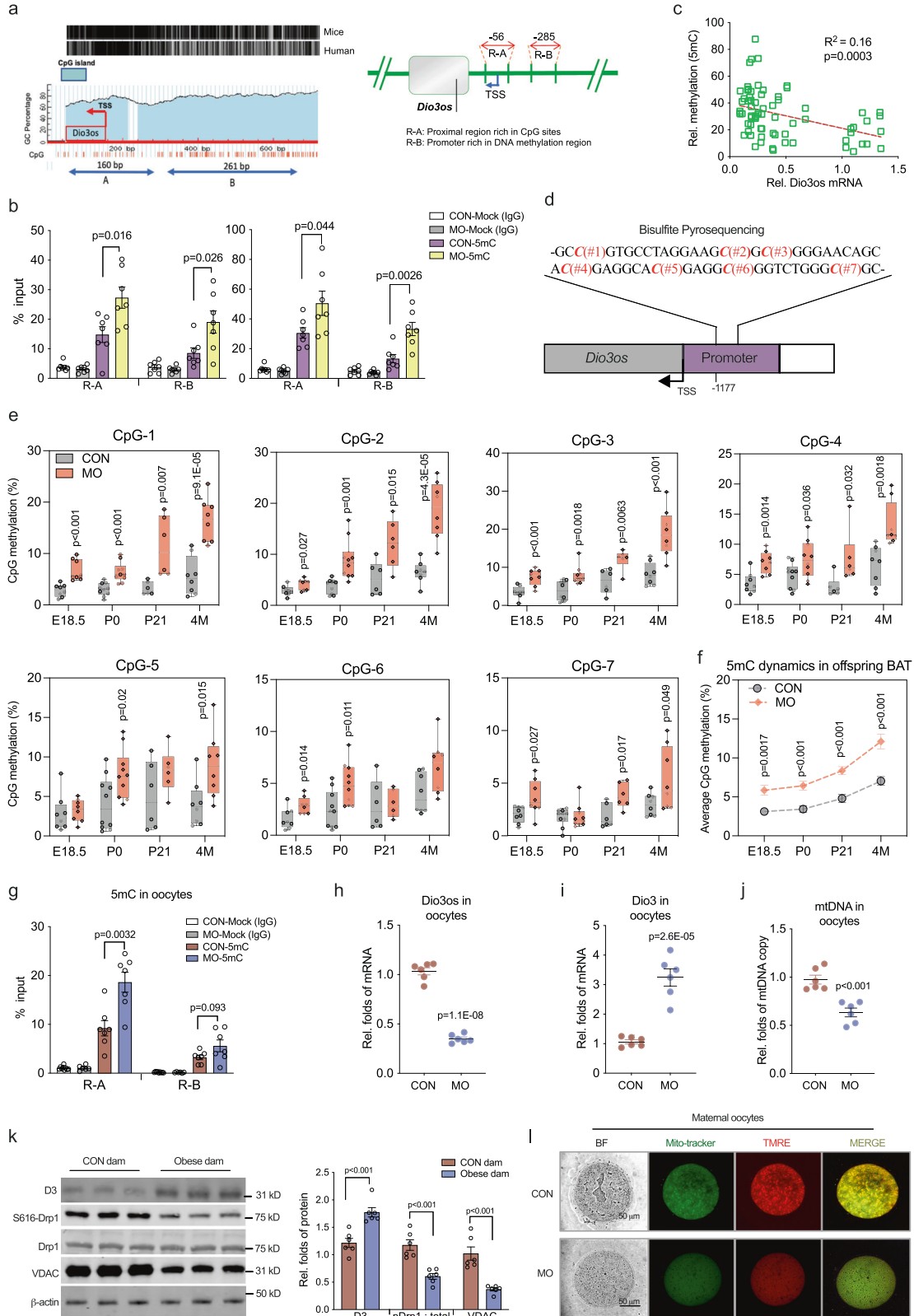

with the essential role of thyroid hormone for oocyte mitochondrial biogenesis[44,45], mitochondrial fission protein Drp1-S616 and mitochondrial membrane protein VDAC were decreased in obese dam oocytes (Fig. 8k), which aligned with the reduced mitochondrial density and mitochondrial membrane potential (Fig. 8l and Supplemental Fig. 12i). Taken together, these data revealed that MO increases promoter methylation of imprinted lncRNA *Dio3os* in oocytes, which persists in fetal and offspring BAT, potentially linking *Dio3os* suppression to the programming of metabolic dysfunctions in offspring.

**Fig. 8 Maternal obesity (MO) increases DNA methylation in *Dio3os* promoter in oocytes and offspring brown fat (BAT). a** Diagram displays proximal region (R-A) and promoter (R-B) in *Dio3os* with enriched CpG island. The region is highly conserved between the human and mouse genomes. **b** Enrichment of 5-methylcytosine (5mC) quantified by MeDIP-qPCR in *Dio3os* proximal region R-A and promoter R-B with enriched CpG islands in fetal BAT at E18 and P0. Mock IgG was used as a negative control ($n = 7$). **c** Regression of *Dio3os* DNA methylation with *Dio3os* expression in fetal BAT. Statistical analyses were performed by Pearson correlation. **d, e** Diagram of bisulfite pyrosequencing to assay percentage of CpG methylation in the *Dio3os* promoter (**d**). CpG methylation percentage in the *Dio3os* promoter in offspring BAT at E18.5 ($n = 8$), birth ($n = 10$), weaning ($n = 6$) and 4-month-old ($n = 8$) (**e**). Whiskers of box plots show mean and individual values from minimum to maximum. **f** Dynamics of CpG methylation percentage in *Dio3os* promoter in offspring BAT measured by bisulfite pyrosequencing ($n = 8$ at E18.5, $n = 10$ at P0, $n = 6$ at P21, $n = 8$ at 4-month-old). **g** Enrichment of DNA methylation (5mC, % input) in *Dio3os* proximal CpG rich region R-A and promoter R-B region in oocytes of control and obese dam measured by MeDIP-qPCR (350–400 pooled oocytes, $n = 7$). Mock IgG was used as a negative control. **h, i** mRNA expression of *Dio3os* and *Dio3* in dam oocytes (150 pooled oocytes, $n = 6$). **j** Mitochondrial DNA (mtDNA) copy in maternal oocytes (25–30 oocytes, $n = 6$). Amplification of mitochondrial genes was normalized to GAPDH and 18 S rRNA. **k** Immunoblotting measurement of DIO3, DRP1, DRP1 phosphorylation at S616 and VDAC protein contents in oocytes (250 pooled oocytes, $n = 6$). β-actin was used as a loading control. **l** Mitochondria (mito-tracker; green color) and mitochondrial membrane potentials (TMRE; red color) in oocytes. Signal intensity was qualified by Image-J (four oocytes, $n = 6$ mice). Data are mean ± s.e.m. and each pregnancy (dam) was treated as replicate unit; statistical analysis was conducted with the unpaired Student's $t$ test with the two-tailed distribution.

## Discussion

Early development of fetuses defines tissue/organ developmental trajectory, causing long-term impacts on offspring health[46,47]. MO predisposes offspring to obesity and metabolic dysfunctions, which form a vicious cycle perpetuating current obesity epidemics[6]. BAT thermogenesis dissipates excessive dietary energy, preventing hypothermia and obesity[11,12]. MO increases ectopic lipid deposition in the liver, WAT, and skeletal muscle of offspring[48,49], but its impacts on the energy expenditure of thermogenic fat remain undefined. Our data showed that MO profoundly impaired female offspring's BAT development and thermogenic function, contributing to reduced glucose and insulin sensitivity and adiposity. We further revealed that BAT of MO female offspring had T3 deficiency, which was associated with *Dio3os* inactivation, a maternally imprinted lncRNA[19,33,34]. *Dio3os* inactivation in BAT hampered brown adipocyte differentiation and thermogenesis, whereas *Dio3os* activation increased PRDM16 activity and thermogenesis, promoting whole-body energy expenditure. Consistently, recovering *Dio3os* expression in MO female offspring profoundly increases BAT thermogenesis, preventing offspring from HFD-induced obesity and metabolic dysfunctions at ambient temperatures. *Dio3os* is a maternally imprinted gene protected from genome-wide DNA methylation during fertilization and early embryonic development[20,21,50]. Consistently, *Dio3os* promoter had higher DNA methylation in oocytes of the obese dam, which persisted in offspring BAT, revealing an epigenetic modification in contributing to intergenerational obesity (Fig. 9).

Brown adipocytes are differentiated from brown precursors in BAT[9,51]. These precursors mediate BAT growth and thermogenic plasticity in response to cold and hormone stimulus[9]. Brown adipogenesis is highly active during the perinatal stage[7,10]. MO impaired BAT lineage determination in fetuses, leading to a persistent decrease of brown precursors in offspring BAT and contributing to reduced energy expenditure and cold intolerance. The lower energy expenditure and metabolic dysfunction of MO offspring were blunted under TN, underscoring the importance of BAT in mediating metabolic dysfunction of MO offspring under cold exposure.

In this study, MO attributing to offspring energy metabolic dysfunctions mainly in females, not males; this gender dimorphism in response to MO has been reported in the previous studies[5,22]. Similarly, the fetal programming of maternal diabetes is mostly transmitted through the maternal line across multiple generations[52]. Despite molecular control for this sexual difference remains obscure, testosterone and estrogen may differentially alter BAT thermogenesis[53], and drive sex- and tissue-related difference in epigenomic memory owing to obesity[54,55]. Meanwhile, sexual antagonism theory suggests that female offspring favor inheriting an epigenetic and transcriptional pattern of maternal imprinted genes[56,57], which might also contribute to *Dio3os* inactivation mainly in females. Thyroid hormone is required for BAT development, mitochondrial biogenesis, and uncoupling thermogenesis[13,14,16]. Meanwhile, thyroid hormone deficiency in the fetus impairs brown adipogenesis, leading to impaired postnatal thermogenesis and oxidative phosphorylation[16]. We found that MO reduced T3 concentration while aberrantly inactivating *Dio3os* expression in female offspring BAT. We further found that *Dio3os* inactivation reduced intracellular T3 and PRDM16 activity, impairing brown adipogenesis and thermogenesis. Consistently, *Dio3os* activation in BAT increased cellular T3 content and PRDM16 activity, which promoted BAT growth and energy expenditure. However, *Dio3os*-induced brown thermogenic activation was blocked by ectopic *Dio3* expression, highlighting *Dio3* silence as a dominant mechanism. Aligned with BAT thermogenic activation following AAV administration to recover *Dio3os* expression, energy expenditure, glucose tolerance, and lipid deposition in WAT and liver were improved in MO female offspring, showing *Dio3os* activation in BAT as a promising target to intervene intergenerational obesity and metabolic diseases.

Imprinted genes emerge as key regulators for mammalian metabolic and physiological adaptations to the environment, such as cold[19,20]. *Dio3os* is located in the *Dlk1-Dio3* imprinting locus and dominantly expressed from a maternally inherited allele, while protein-coding gene *Dio3* is expressed from paternally inherited allele[17,18,33,58]. *Dio3os* suppression promoted *Dio3* expression and attenuated BAT function, which supports the kinship theory that maternally expressed genes favor higher levels of thermogenic activity but paternally expressed genes tend to reduce thermogenic outputs of individuals[19]. Supportively, a number of human genetic diseases are consequences of inappropriate alterations of imprinted genes, such as Prader-Willi, Angleman, and Silver-Russel syndromes[20]. Our study revealed that epigenetic dysregulation of imprinted lncRNA *Dio3os* is also associated with intergenerational obesity and metabolic dysfunction in females. Recently, lncRNA *Dio3os* was reported to promote the proliferation of human cells[59,60], which is mediated by direct interaction with miRNAs, such as miR-122[59]. Because miR-122 activation is associated with obesity[61], whether miR-122 mediates *Dio3* suppression by *Dio3os* warrants further studies.

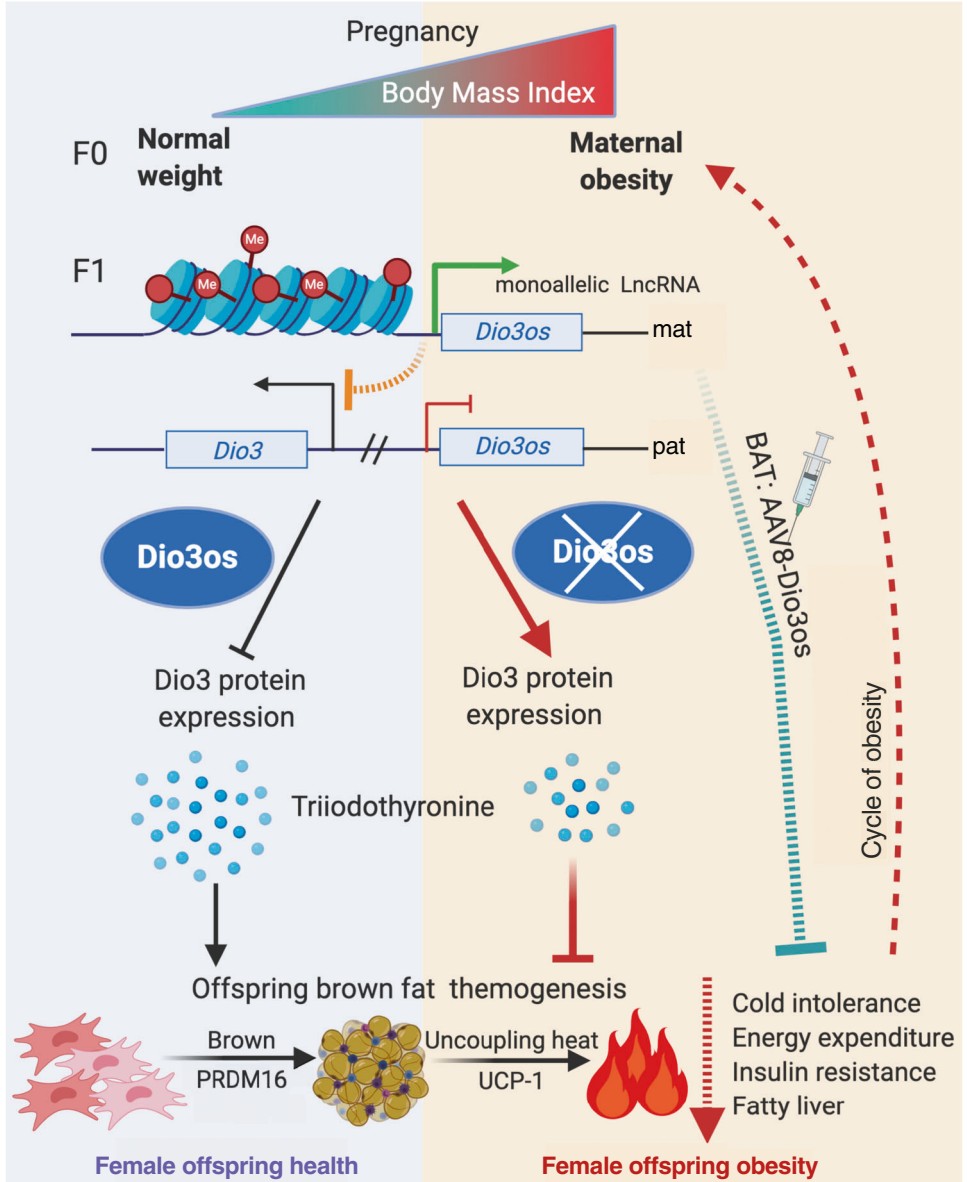

**Fig. 9 Diagram summaries that maternal obesity impairs thermogenesis and energy expenditure of brown adipose tissue (BAT), predisposing female offspring to obesity and metabolic dysfunctions.** *Dio3os*, a maternally imprinted long-coding RNA, has DNA hypermethylation in its promoter, which has an oocyte origin. The hypermethylation attributes to *Dio3os* inactivation in offspring BAT, which activates Dio3 and reduces thyroid hormone T3 action, leading to impaired PRDM16 activity and thermogenesis. Mouse BAT receiving AAV vector expressing *Dio3os* prevents MO female offspring from diet-induced obesity and glucose intolerance, providing a pharmaceutic opportunity to improve the metabolic health of offspring and intervene in a vicious maternal-daughter obesity cycle.

Differential DNA methylation is required for gene imprinting[20]. We found that DNA methylation was increased in the *Dio3os* promoter of obese dam oocytes, and reduction of thyroid hormone signaling has been reported to impair mitochondrial biogenesis in oocytes[62]. Altered DNA methylation patterns of imprinted genes have been also reported in oocytes of patients with type 2 diabetes though the biological consequences of such changes remained unclear[63]. During gametogenesis and early embryonic development, although genome-wide DNA methylation erasure occurs in non-imprinted gene promoters, methylation in imprinted genes escapes, which leaves footprints in offspring[21]. Persistent DNA hypermethylation was discovered in the *Dio3os* promoter of obese dam oocytes, fetal and offspring BAT, which

potentially programmed *Dio3os* inactivation and metabolic dysfunctions in offspring.

In summary, we find that MO epigenetically regulates the expression of lncRNA *Dio3os*, which promotes *Dio3* expression, and suppresses intracellular T3 level and fetal BAT development, as well as female offspring BAT thermogenic function. *Dio3os* activation increases BAT thermogenesis, improves insulin sensitivity and energy expenditure under both cold and TN conditions. These findings deepen our understanding of the programming effects of MO on thermogenic energy expenditure in offspring, providing a mechanism explaining developmental origins of offspring obesity and metabolic dysfunctions. *Dio3os* activation in BAT also provides a therapeutic target for manipulating BAT thermogenic activity to prevent obesity.

## Methods

**Animal and diets**. Wild-type female C57BL/6J mice at 8 weeks of age (The Jackson Laboratory, Bar Harbor, ME, USA) were randomly separated into two groups and fed ad libitum either a control diet (10% energy from fat; D12450H, Research Diets, New Brunswick, NJ, USA) or HFD (45% energy from fat; D12451, Research Diets) for 10 weeks to induce obesity. Then, females were mated with males fed a regular chow diet. Successful mating was determined by the presence of a copulation plug in the vagina. After mating, female mice were maintained on their respective diets until pups were weaned at postnatal day 21 (P21). At E18, mice were killed, and fetuses and fetal BAT were collected and weighed. Owing to the small size of fetal BAT, fetus from the same little was pooled. Each dam (pregnancy) was considered as an experimental unit. At birth, the litter size was standardized to six. After weaning, female and male offspring from control and MO were further randomly assigned to either a control diet (10% energy from fat) or HFD (60% energy from fat, D12492, Research diets) for 12 weeks at 22 °C in 12-h light/12-h dark and 50% humidity facility. In addition, a subset of 4-month-old offspring was subjected to 30 °C for 4 weeks or 4 °C for 4 days to test BAT independent effects. Offspring interscapular BAT and neonatal skin in the dorsal region were collected for analyses.

For germinal vesicle stage oocyte collection, ovaries were harvested and placed in Dulbecco's Modified Eagle Medium (DMEM) free fetal bovine serum (FBS) medium. Large antral follicles were punctured with a gauge needle, and oocytes were removed through a glass pipette. All animals were euthanized by $CO_2$ inhalation and cervical dislocation. All animal studies were conducted according to the protocol approved by the Institute of Animal Care and Use Committee (IACUC; ASAF#6704) at Washington State University (WSU, WA, USA). The animal facility is accredited by the Association for Assessment and Accreditation of Laboratory Animal Care (AAALAC).

**Cell culture and brown adipogenic induction**. MEFs were isolated from wild-type mice at E13.5 as previously described[51]. In brief, the fetal head, front and back limb, and the internal organs were removed, and the remaining somatic tissues were rinsed and minced. Tissues were digested in 0.025% trypsin/ethylenediaminetetraacetic acid (EDTA) (Invitrogen, CA) for 30 min in a shaking incubator at 37 °C. The cell lysate was filtered through 40 µm strainers. After centrifugation at $400 \times g$ for 5 min, cells were suspended in DMEM supplemented 10% FBS and 1% penicillin-streptomycin. MEFs were grown to 80% of confluence and followed with brown adipogenic induction with DMEM (10% FBS) supplemented with an adipogenic cocktail, including 0.1 µg/ml insulin, 0.5 mM isobutylmethylxanthine, 1 µM dexamethasone, 125 µM indomethacin, and 1 µM T3[51,64]. Cells were refed every 48 h with a maintenance cocktail containing 0.1 µg/ml insulin and 1 nM T3. Cells were fully differentiated by day 6 as shown by the presence of lipid droplets when observed under an optical microscope.

**Spheroid culturing and brown adipogenic induction**. BAT-SVFs were induced to form spheroids as previously described[39,40]. For isolating SVFs, BAT was dissected and digested in DMEM FBS free medium containing 0.75 U/ml collagenase D (Roche, Pleasanton, CA) for 30 min in a shaker at 37 °C. The lysate was filtered through 100 and 40 µm strainers. Cells were collected by centrifuge at $500 \times g$ for 5 min. Isolated SVFs were suspended in EGM2 containing 10% FBS and seeded in ULA plates for 3 days. Spheroids were pipetted and incorporated in 100 µl reduced-growth factor Matrigel (Corning; New York, USA) and seeded into attached plates containing EGM2. After 5 days of culturing to allow cells migration into Matrigel, cells were further differentiated in DMEM medium containing adipogenic cocktail composed of 0.1 µg/ml insulin, 0.5 mM isobutylmethylxanthine, 1 µM dexamethasone, 125 µM indomethacin, and 1 µM T3 for 5 days. Cells were differentiated into a maturation medium containing 0.1 µg/ml insulin and 1 µM T3 for additional 20 days[65,66].

**Adenovirus-associated virus (AAV) preparation and injection**. AAV virus particles with serotype AAV8 were generated by GeneCopoeia Medical Center (Rockville, MD, USA)[67]. To generate AAV-Dio3os plasmid, the DNA fragment coding for Dio3os was extracted by SalI and NedI digestion from pUC-Dio3os, and subcloned into a pAAV2 cloning plasmid using the same restriction sites. pAAV-Dio3os plasmid contains inverted terminal repeats (ITRs) and Dio3os cDNA downstream of a CMV promoter, which can induce efficient transduction in BAT[68]. To produce AAV viral particles with serotype AAV8, pAAV-Dio3os vector was further co-transfected into AAV-293 cells with AAV8 cap genes and adenoviral helper genes for producing recombinant AAV8-Dio3os-CMV viral particles. After 72 h, the supernatant was collected and centrifuged at $1500 \times g$ for 15 min, and the pellet was resuspended in lysis buffer and kept at −80 °C. Viral particles were further purified by ultracentrifugation at $350,000 \times g$ in an iodixanol gradient following concentration using Amicon Ultra-15 (Millipore). Viral particle titers ranged from $1 \times 10^{13}$ to $3 \times 10^{13}$ genome copies (GC)/ml. Approximate $1.5 \times 10^{11}$ and $5 \times 10^{11}$ GC viral particles were individually injected into interscapular BAT of neonatal and adult mice as previously described[41,68].

**CRISPR-cas9 and plasmid transduction**. To knockdown Dio3os expression, CRISPR-Cas9 RNA (crRNA) targeting Dio3os or non-targeting scramble were delivered into cells by electroporation using the Neon Transfection System (Invitrogen, Carlsbad, CA) according to manufacturer's instructions[65,66]. To form the ribonucleoprotein (RNP) complex, dio3os crRNA (IDT, Iowa, USA), Alt-R CRISPR-tracrRNA complex (IDT) and Cas9-N-NLS nuclease (GeneScript, NJ, USA) were mixed and incubated at room temperature for 15 min. For RNP electrophoresis, $1 \times 10^5$ cells were digested with 0.25% trypsin. Washed cells were suspended in 5 µl RNP and 3 ml Neon R electroporation buffer by gentle pipetting. The mixture was transferred to a sterile cuvette and electroporated with 1350 V, 30 ms width, two pulses. Cells were studied 71 h after transfection. crRNA sequences are provided in Supplementary Table 1. Plasmid Dio3 ORF (NM_172119.2) was purchased from GeneScript (Piscataway, NJ, USA). Plasmid transfection was performed using lipofectamine 3000 transfection reagent (Invitrogen) according to manufacturer's protocol.

**Indirect calorimetry**. Indirect open circuit calorimetry measurements were performed with the Comprehensive Lab Animal Monitoring System (CLAMS; Columbus Instruments, Columbus, OH, USA) as previously described[65,66]. Mice were singly caged and acclimated to the metabolic chamber before metabolic recordings were conducted by Oxymax v 4.93 (Columbus Instruments). During the measurement, mice were maintained on their respective diets and water ad libitum. Data were normalized according to ANCOVA energy expenditure guideline[23].

**Temperature measurement**. The surface temperature was measured with E6 infrared thermal camera (FLIR system, OR, USA) and images were analyzed with FLIR-Tools-Software (FLIR system)[51,65,66]. After mice were separated from nests, the surface temperature was immediately obtained in order to prevent the noise from ambient temperature and heat loss. In capturing imaging, distance and animal behavior were also controlled to reduce possible artificial effects. To reduce thermal variations, rectal body temperature was also measured using a highly precise electronic thermometer (Thermalert TH-5, Physitemp Instrument).

**Glucose and insulin tolerance tests**. Mice fasted for 6 h with water provided. Mice were intraperitoneally injected with 2 g/kg of body weight of D-glucose or 0.5 U/kg body weight of insulin. Glucose was measured in the serum obtained from the tip of the tail at 0, 15, 30, 60, and 120 min after administration. The concentration of glucose was measured using an automatic glucose monitor (Contour, Bayer, IN, USA)[65,66]. Maternal insulin concentration in serum was measured using the mouse insulin enzyme-linked immunosorbent assay (Alpco, NH, USA). Insulin resistance (HOMA-IR) was calculated according to the formula: HOMA-IR = ( fasting blood glucose mg/dl × 0.055) × (fasting insulin µU/ml) / 22.5[65,66].

**Methylated DNA immunoprecipitation (MeDIP)**. MeDIP analysis was conducted as previously described[51,65]. In brief, brown adipose tissues and oocytes were digested in tail lysis buffer (20 mM Tris-HCl, 4 mM EDTA, 20 mM NaCl, 1% SDS) and protease K overnight in water bath at 55 °C. Tissue DNA was isolated using Tris-phenol, chloroform, and isoamyl alcohol. Isolated DNA was dissolved into TE buffer and sonicated to produce 300–500 bp fragments (30% power, 10 s on/off 5 min), which was verified by electrophoresis on a 2% agarose gel. Two micrograms of denatured DNA were incubated with 2 µg antibodies against 5-methylcytosine (5mC) (# A3002; ZYMO Research) or anti-mouse IgG (#31430; Thermo; 1:300) overnight. DNA-5mC complexes were pulled down with pre-washed protein A magnetic beads (#73778; Cell signaling) at 4 °C for 1 h. Captured beads were washed three times with buffer (0.1% SDS, 1% Triton X-100, 20 mM Tris-HCl pH 8.1, 2 mM EDTA, 150 mM NaCl) and twice with TE buffer. DNA was recovered in digestion buffer (50 mM Tris-HCl pH 8.0, 10 mM EDTA, 0.5% SDS, 35 µg proteinase K) and incubated for 3 h at 65 °C. Recovered DNA was used for qPCR analysis. Primers for MeDIP-qPCR are listed in Supplementary Table 1.

**Bisulfite pyrosequencing**. Pyrosequencing and data analysis were performed as previously described[65]. Briefly, genomic DNA was converted by bisulfite using DNA Methylation-Direct Kit (D5021; Zymo Research). Converted genomic DNA was used as a template to amplify the target sequence. GC-purified biotinylated primers were targeted to the Dio3os promoter or Dlk1-Dio3 IG-DMR.

**Immunocytochemistry and Oil-Red O staining**. BAT was fixed in 4% paraformaldehyde (PFA) for 24 h at room temperature and embedded in paraffin. Paraffin sections (5 µM thickness) were subjected to immunochemical staining. Tissues were incubated in cold 4% PFA for 10 min and permeabilized with 0.25% Trition X-100 for 10 min. After blocking with 1% BSA for 30 min, cells were incubated with a primary antibody anti-rabbit UCP-1 (1:200; #14670; cell signaling) at 4 °C overnight. After washing for three times with PBS, a fluorescent anti-rabbit second antibody (1:1000, Alex-Fluor-555-Red; Biolegend, CA, USA) or 1 mM MitoSpy-green (#424805; Biolegend) or CTB (#C34775, Alex-Fluor-488-Green, Thermo Fisher) was added and protected from light for 1 h at room temperature. The background was determined using the same procedure but substituting primary antibodies with a general mouse IgG to decrease nonspecific background. For mitochondrial membrane potential measurement, oocytes were stained by TMRE-red (#115532; Sigma). Images were obtained with an EVOS XL

Core imaging system (Mil Creek, WA, USA) and Lecia confocal laser SP5. Lipids were stained with Oil-Red O. Briefly, cells were fixed in 4% PFA for 10 min. After washing, cells were stained by Oil-Red O in 60% isopropanol. In addition, Oil-red O was dissolved into 100% isopropanol, and absorbance was measured at 510 nm in a Synergy H1 microplate reader (Biotek, Winooski, Vermont, USA)[51,65].

**Real-time PCR**. Total RNA in tissues or cells was isolated by TRIzol reagent (Invitrogen, NY, USA) following the manufacturer's guidelines. mRNA was reversed to cDNA by iScript[TM] cDNA synthesis kit (Bio-Rad, CA, USA). iQ[TM] SYBR Green Supermix (Bio-Rad) system was used for quantitative real-time PCR (IQ5, Bio-Rad), and data were collected by Bio-Rad CFX manager 3.1 software (CFX Connect)mRNA expression was normalized to 18 S rRNA. Primer sequences are listed in Supplementary Table 1.

**Electron microscopy**. Electron microscopic imaging was processed as previously described[51,65]. In brief, fresh BAT was cut into small pieces and fixed in a solution containing 4% PFA (EM grade) and 3% glutaraldehyde overnight at 4 °C, followed by secondary microwave fixation at 250 w for 2.5 min. Samples were stored overnight in 2% OsO4 at 4 °C and then processed by gradient ethanol dehydration at room temperature. Samples were then stored in propylenoxid: resin (1: 1, v/v) and 3× in resin at room temperature overnight. Then, samples were immersed in fresh resin overnight at 62 °C. After embedding, semi-thin sections (1 μm) were cut with a Lecia ultramicrotome, and ultra-thin sections (60 nm) were produced by the presence of silver color shadow. The sections were fitted to grids and incubated in uranylacetate and stained with lead citrate. Samples were imaged by transmission electron microscope (FEI Technai G2 20 Twin (200 kv LaB6, OR, USA).

**Flow cytometry**. Flow cytometry was performed as previously described[51,65]. BAT primary SVFs were freshly prepared and fixed in 4% PFA for 15 min. Fixed cells were slowly permeabilized with 100% methanol for 15 min on ice. Washed cells were blocked in 1:300 conjugated PDGFRα APC (1:200) (#135907; Biolegend, USA) and EBF2 anti-rabbit antibody 1: 200 (#bs-11740R; Bioss, MA, USA) at room temperature for 2 h. Cells were further incubated in Alex-Fluor-anti-rabbit-488 (#100343867; Biolegend) for 1 h and sorted in a Sony SY3200 sorter (Sony, CA, USA). FACS data were analyzed by FlowJo software (Treestar Inc., San Carlos, CA). Gates were determined based on fluorescence minus control.

**Immunoblotting**. Protein samples were isolated from tissue or cells with lysis buffer[51,65]. Homogenized protein lysates were quantified using a BCA protein assay kit (#K819; BioVision, USA). For immunoblotting, proteins were separated by sodium dodecyl sulfate–polyacrylamide gel electrophoresis on 10% polyacrylamide gels and transferred onto a nitrocellulose membrane. Membranes were probed using primary antibodies, including anti-UCP-1 (#14670; Cell Signaling; 1:1000), PRDM16 (PA5-20872; Thermo Fisher Scientific; 1:1000), PGC-1a (#66369-I; Proteintech; 1:1500), Dio2 (ab77481; Abcam; 1:1000), Dio3 (ab233034; Abcam; 1:1000), VDAC (#4866; Cell Signaling; 1:1000), CtBP1 (ab14411; Abcam; 1:1000), TH (25859-1-AP; ProteinTech; 1:1000), β-actin (MA5-15739; Thermo Fisher Scientific; 1:1500) or β-tubulin (#179513; Abcam; 1:1500). Secondary antibodies, including anti-mouse IRDye 680 (C80926-17; 1:10,000) and anti-rabbit IRDye 800CW (C70918-03; 1:10,000), were purchased from LI-COR Bioscience (Lincoln, NE, USA). Signals were detected using infrared imaging (Odyssey, LI-COR Biosciences). Band intensity was quantified using Image Studio Lite 5.2 (Licor Biosciences).

**Co-immunoprecipitation assay**. Cell lysate (0.75% NP-40, 1 mM DTT) containing protease inhibitors (#87786; Thermo Fisher) was precleared with protein A magnetic beads (#73778; Cell Signaling) at 4 °C for 1 h[51,65]. 2 μg of primary antibody anti-PRDM16 (PA5-20872; Abcam) was added to lysate containing 500 μg total protein and mixed by rotation overnight at 4 °C. Then, protein A magnetic beads were added to the precleaned solution and gently rotated for 4 h at room temperature. Beads were washed with cold PBS three times and collected for immunoblotting.

**T3 and T4 in BAT**. Fresh BAT was minced and homogenized in cold 100% methanol, chloroform, and barbital buffer with a mixture of 1:4:2 (v/v) to extract thyroid hormones as previously described[69,70]. Tissue samples were centrifuged at 1500 × g for 10 min at 4 °C, and the supernatant was removed and assayed with ELISA (#8025-300; Monobind Inc, CA, USA). T3 and T4 concentrations were normalized to the protein content in BAT (BCA assay, BioVision, USA). Serum concentrations of T3, T4, and TSH were assayed and analyzed by BioTek Synergy H1 (Gen5 2.01 software) according to the manufacturer's guidelines.

**Oxygen consumption rate (OCR)**. Extracellular oxygen consumption was measured according to an assay protocol (ab197243, Abcam). Inguinal-WAT and liver were freshly prepared, weighed, and cultured in DMEM at 37 °C. Mineral oil was added to limit the diffusion of oxygen into the assay medium. Fluorescent intensity was measured by a time-lapse microreader at 360 nm excitation and 680 nm emission. OCR data were normalized to tissue mass.

**Epinephrine and norepinephrine**. Epinephrine and norepinephrine in serum were measured by ELISA assays (KA1877, Abnova, CA, USA) and analyzed by BioTek Synergy H1 (Gen5 2.01 software) according to the manufacturer's guideline.

**Data presentation and statistical analyses**. Data are presented as mean ± s.e.m. All statistical analyses were performed using SAS version 9.4 (SAS Institute, NC, USA) and figures were prepared using GraphPad Prism7 (San Diego, CA, USA). Each pregnancy (dam) was considered as a replicate unit for offspring data analyses. Unpaired two-tail Student's $t$ test, one- or two-way analysis of variance (ANOVA) with adjusted-comparison was used in data analyses. Feed intake data were measured by ANOVA repeated measurements. Metabolic chamber data were analyzed by NIDDK MCCP ANCOVA multiple linear regression models. Significant differences are indicated as *$P < 0.05$, **$P < 0.01$, and *** $P < 0.001$.

**Reporting summary**. Further information on research design is available in the Nature Research Reporting Summary linked to this article.

## Data availability
The authors declare that the data supporting the findings of this study are available within the paper and its supplementary information files. Source data are provided with this paper.

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

## Acknowledgements

This work was funded by the National Institute of Health R01-HD067449 (to M.D.) and R21-AG049976 (to M.D.). The authors thank Franceschi Microscopy & Imaging Center for help in electronic microscopy imaging.

## Author contributions

Y.T.C., M.D. developed the concept, designed experiments, and Y.T.C., P.W.N, M.D. interpreted the data. Y.T.C., Y.H., X.D.L., J.M.A. conducted experiments and collected data. Y.T.C. and Q.Y.Y. analyzed data. Y.T.C., P.W.N., M.D. prepared the manuscript. Y.T.C., M.D., P.W.N. and Z.M.J. made revisions to the manuscript. All authors approved the final content. Y.T.C. and M.D. are the guarantors of this work and had full access to the data in the study and take responsibility for the integrity of the data and accuracy of the data analysis.

## Competing interests

The authors declare no competing interests.

**Additional information**

**Peer review information** *Nature Communications* thanks Adilson Guilherme, Andrea Riccio, and the other anonymous reviewer(s) for their contribution to the peer review this work. Peer reviewer reports are available.

