## [Peer Review File · Nature Communications]

Reviewers' Comments:

Reviewer #1:

Remarks to the Author:

Manuscript: NCOMMS-21-04369 – “Imprinted LncRNA Dio3os preprograms intergenerational brown fat thermogenesis and obese resistance”. Chen Y-T et al.,

In this manuscript, Chen and colleagues investigated how maternal obesity (MO) predispose the offspring to obesity and metabolic dysfunctions. The authors found that MO inhibits the expression of a long noncoding Dio3 antisense RNA (Dio3os), resulting in enhanced expression of the Dio3 enzyme and reduction of intracellular T3 levels. In turns, the T3 deficiency impairs brown fat cell differentiation and fetal BAT development, favoring metabolic dysregulation and insulin resistance in MO female offspring.

Based on these new findings, the authors proposed that MO impairs female offspring BAT development and thermogenic function, contributing to insulin resistance and glucose intolerance.

The major claims are:

1. Maternal obesity (MO) renders female offspring metabolic dysfunction and cold sensitivity due to BAT dysregulation.
2. Impaired brown adipogenesis in MO female offspring has a fetal origin.
3. MO dysregulates maternally imprinted Dio3-Dio3os in fetal BAT, enhances Dio3 enzyme expression causing T3 deficiency.
4. Dio3os inactivation blocks brown adipogenic differentiation
5. Conversely, Dio3os activation in brown fat increases PRDM16 activity and body energy expenditure in female mice.
6. Dio3os activation in BAT prevents MO female offspring from diet-induced obesity
7. Dio3os hypermethylation mediates Dio3os suppression in obese dam oocytes and female offspring BAT.
8. The authors concluded that maternal obesity epigenetically regulates the expression of imprinted LncRNA Dio3os, which suppresses fetal BAT development and BAT functions.

Critique:

Overall, this is a well-written manuscript, presenting exciting new results and solid conclusions. However, there are some issues and questions that need to be addressed before the paper merit publication in Nature Communications (below):

Other Points:

1. The authors state that MO enhances the expression of deiodinase-3 (Dio3) and leads to intracellular T3 deficiency and impaired fetal BAT development. Due to the importance of thyroid hormones in different tissues' development and function, it would be critical to examine whether MO also enhances Dio3os methylation, Dio3 expression and causes T3 deficiency in other T3-regulated tissues such as the liver, skeletal muscle, heart, nerve.
2. Importantly, the authors have not addressed whether MO can also affect the browning of white adipose tissue (WAT) via a similar mechanism, such as T3 deficiency. Several reports indicating that WAT browning improves systemic insulin sensitivity and glucose metabolism have been published over the last decade. Thus, it would be essential to examine if MO affects Dio3 expression, T3 levels, adipogenesis, and WAT browning induced by cold.
3. Also, as the T3 is essential for the nervous system function, the authors need to examine whether the MO-induced imprinting of LncRNA Dio3os affects the local production of T3 in other tissues besides BAT. Particularly the nerve system, which the normal development requires optimal levels of T4/T3. Moreover, abnormal sympathetic innervation and outflow in adipose tissue depots may impact BAT and WAT development, increasing the susceptibility to metabolic disorders.

4. Although the results in Figure 2 suggest BAT's importance in mediating the metabolic dysfunction of offspring MO female, it would be interesting to test whether AAV-mediated overexpression of Dio3os (AAV-Dio3os) can activate BAT even at thermoneutrality to improve systemic metabolism in MO offspring mice.
5. In Figure 3F and page 8 (line 164), the authors state that MO attenuated T3 levels in fetal and neonatal BAT. However, the information related to T3 levels in adipose tissue depots (BAT and WAT) from MO adult mice is missing. This info needs to be included.
6. Figure 4 and Figure S7 would be important to determine whether the binding of PGC1a to PRDM16 is also inhibited in MO adult female mice.
7. In Figure 6, the UCP1 expression in BAT from MO female mice was not reduced compared to control. Moreover, AAV-Dio3os administration did not seem to affect much UCP1 expression in BAT from MO mice. The quantification of UCP1 and other thermogenic protein levels in control, MO, PBS-treated versus AAV-Dio3os-treated mice should be conducted.
8. As the higher methylation of Dio3os promoter and reduced Dio3os expression was also observed in adult BAT (Figure 7d), the authors should examine thyroid hormone levels and THR signaling in BAT from MO adult mice.
9. The authors state that recovering Dio3os expression in MO female offspring increases BAT thermogenesis, preventing offspring from western diet-induced obesity and metabolic dysfunctions. However, they did not investigate the effect of AAV-Dio3os-OE on BAT expansion in prolonged cold exposure (i.e., 4 days at 4°C, as shown in Figure 2).

Reviewer #2:

Remarks to the Author:

The prevalence of obesity is increasing, even among women of childbearing age. In addition to the short-term complications of obesity for both mother and child, emerging evidence suggests that maternal obesity has long-term detrimental consequences for offspring health. Though multiple genome-wide association studies have been carried out in the past decades to try to identify the genetic basis of these disorders, the factors responsible for these diseases and their heritability are still obscure. Furthermore, nongenetic components, specifically epigenetic factors, has recently been recognized, and may also contribute to offspring traits acquired. In this manuscript, the author reports maternal obesogenic diet before mating and during gestation renders female offspring metabolic dysfunction and cold intolerance, and they showed lower BAT mass and downregulated genes involved in brown adipogenesis in MO female offspring, and they were originated from fetal stage. Mechanistically, enhanced expression of Dio3 inactive T3 to T2, which leads to intracellular T3 deficiency and suppression of BAT development. To show the regulatory role of Dio3os for Dio3 expression, the authors provide the evidence: *in vitro* knockdown of Dio3os activated Dio3 expression, reduced intracellular T3 availability and expression of Thra/Thrb, and blocked brown adipogenic differentiation; *in vivo* overexpression of Dio3os reduced D3 content, increased BAT mass, and activated BAT thermogenesis, which consequently prevented MO female from western diet induced obesity. Furthermore, they show high methylation of Dio3os promoter contributes to the reduced Dio3os expression in MO offspring BAT. The authors propose that methylation changes in fetal and adult BAT were originated from oocytes, and conclude that methylation in Dio3os promoter may serve as a novel imprinted lncRNA in mediating intergenerational obesity.

This study is performed on the MO induced fetal BAT phenotypic analysis, and presents the evidence to support Dio3 dysregulation mediates maternal obesity induced fetal BAT development defects. Dio3 function has been well recognized. Some findings, for example, Dio3os suppresses Dio3 expression, and its expression is controlled by methylation in promoter region that behaves like a maternal imprinted gene, are interesting. However, the molecular mechanism underlying this process has not been clearly presented here, which is important for people to understand the intergenerational effects of BAT in obesity.

Major concerns:

1. In the study, they showed that brown progenitors and preadipocytes were profoundly reduced in MO fetal BAT, and isolated SVFs in MO fetal BAT had lowered capacity to differentiate into mature brown adipocytes in vitro, consequently decreasing BAT mass. Based on these results, metabolic dysfunction in female offspring is mainly due to impaired brown adipogenesis. Considering that disruption of brown fat thermogenesis is caused by deficiency of mature brown adipocytes, the current title of manuscript may be changed to "...brown fat development or differentiation"?
2. Prdm16 controls a bidirectional cell fate switch between skeletal myoblasts and brown fat cells. Loss of Prdm16 from brown fat precursors results in a loss of brown fat characteristics and promotes muscle differentiation (Nature 2008. 21; 454(7207):961-7). The authors showed that the expression of PRDM16 was suppressed in MO offspring, so for UCP-1 and PGC1-1a protein contents and expression of thermogenic genes. It is better to detect whether the expression of muscle-specific genes are upregulated when PRDM16 is downregulated.
3. In Fig. 1m, expression of genes involved in lipid uptake was decreased in MO offspring BAT; however, in Fig 1I, transmission electron microscopy image showed that lipid contents seem to be accumulated in MO offspring. how to explain that?
4. The authors declare thermogenesis, lipid uptake and oxidization were decreased in MO offspring BAT, contributing to lipid accumulates in WAT and liver (Fig 1n and Fig. 6f). Lipid drop deposited in WAT and liver was due to BAT dysfunction or metabolic defects occur in WAT and liver? To clear this, it is needed to characterize the metabolic traits of WAT and liver.
5. Based on the findings in this report, T3 deficiency was occurred in BAT, not in serum. In supplementary Figure 6c-h, the results showed T4, T3 and TSH concentrations in serum were not significantly altered in neither female fetuses nor neonates. However, this is likely due to the great variance among individuals. Increasing sample size and re-analysis of the data may be needed to clarify this issue.
6. Dio3os overexpression in interscapular BAT activates BAT adipogenesis and thermogenesis, and prevents female offspring from diet induced obesity. Is it possible that AAV-Dio3os vector leaks into circulation and functions in tissues response to thyroid hormone, like WAT and liver?
7. The association between Dio3os inactivation and Dio3 overexpression is extensively examined in this report; however, much of the data is correlative. Dio3 and Dio3os locate in the DLK1-Dio3 imprinting locus. Dio3os, also named Dio3 opposite strand, locates in 1.6 kb upstream of Dio3, expresses from the maternal allele (Development, 2007, 134, 417-426). Dio3, a maternal imprinting gene, expresses from paternal allele. Here, the authors detected the expression of other imprinting genes (both paternally expressed) in the locus, like Dlk1 and Rtl1. How about the expression of other maternal imprinting genes located in this locus, such as Gtl2 and some microRNA? It is also very essential to figure out which allele (maternal or paternal) contributes to the Dio3 expression? Importantly, it is still confusing that how suppression of Dio3os activates the expression of Dio3? By what mechanisms? It is a great shame that these questions are still open in this study.
8. The authors conclude that methylation changes in Dio3os promoter can persist to early embryogenesis and adult tissue, consequently contributing to the metabolic phenotypes in the offspring. Hence, it should not be a tissue specific imprinting. All tissues from MO offspring should be hypermethylated in Dio3os promoter, and the expression of Dio3 and Dio3os may be altered accordingly. In the present study, the authors found that MO male offspring are less susceptible to body weight gain and insulin resistance compared to female. They think this gender dimorphism is resulted from the differential effects of sex hormone. Since Dio3os is maternal imprinting gene as the authors stated in the manuscript, Dio3os promoter in the tissues of male offspring may also acquire the similar methylation pattern, and the expression of Dio3 and Dio3os might be changed. So detection of DNA methylation and gene expression in other female tissues and male BAT would further strengthen the concept of maternal imprinting.

Minor points

1. Some errors in figures: line 108 should be "Fig. 1h and supplemental Fig. 3e,f"; line 110 should be "Fig. 1l and supplemental Fig. 3g";
2. Fig. 2b, BAT mass gain should be presented as relative value.
3. Fig. 2e, the figure was not correctly labeled, 22C?
4. Mitochondrial density in Fig. 1l should be quantified. Similar issues can be found in many figures

in this paper, such as Fig. 3I and other figures (HE and immunofluorescence pictures).
5. Oocyte numbers in Fig. 7 should be included in the Figure legends.

Reviewer #3:

Remarks to the Author:

This study investigates the role of the lncRNA Dio3os in the control of brown fat differentiation and thermogenesis and development of obesity in the mouse. In brief, the authors demonstrated that the female offspring of mice fed with high-fat diet during pregnancy developed obesity and metabolic disorder, which were associated with less BAT and more WAT deposition. Consistent with this finding, this progeny accumulated less BAT and had reduced thermogenic capacity when exposed to cold temperatures. To look for the origin of this metabolic alteration, the authors also found that BAT reduction began at fetal stages and was associated with impaired differentiation capacity of brown fat progenitor cells. The cause of this abnormality was identified in a lower level of the thyroid hormone T3 that in turn was associated with higher expression of the deiodinase Dio3 gene. Because Dio3 is close to the lncRNA gene Dio3os, the authors investigated its expression and found that it was down-regulated in brown fat. By employing a knockdown model in mouse embryonic fibroblasts, they demonstrated that Dio3os negatively regulated Dio3 expression and inhibited in vitro brown adipocyte differentiation that could be rescued by T3 supplementation. Inhibition of Dio3os in a further ex-vivo model obtained with stromal vascular cells isolated from fetal BAT led to similar conclusions. They then forced Dio3os expression by injecting a transgene vehiculated by an adenovirus-based vector into intra-scapular BAT. They found that such treatment decreased Dio3 and increased T3 and brown adipogenesis also reducing the gain of body weight in the mice fed with a normal diet and preventing the development of obesity and metabolic dysfunction in the offspring of obese female fed with a high-fat diet. Finally, they found that Dio3os suppression in fetal BAT of obese dams was associated with the gain of methylation of the Dio3os promoter. Higher Dio3os promoter methylation was found also in adult BAT of the obese mice offsprings and in the oocytes of the obese dams, leading the authors to conclude that maternal methylation persisted in their offspring.

This study provides several new and useful information. However, we think there are some points that can be improved or clarified.

1. I am confused with the different types of high-fat diets used in this study: HFD, western diet, obesogenic diet. Please, clarify the need to use different diets.
2. The issue of gender dimorphism in the development of obesity is interesting but not properly investigated. What is the molecular mechanism causing this gender difference? Is Dio3os downregulated and methylated and Dio3 upregulated also in the male offspring of obese dams? Or is the difference more downstream?
3. I could not find any study demonstrating the imprinting of Dio3os and even imprinting of Dio3 is known to be tissue-specific. Because the authors do not investigate the allele-specific expression of these genes, I think it is improper to refer to their imprinted expression, as they did in several figures. Also, Dio3os promoter methylation could be the consequence rather than the cause of Dio3os repression and the MeDIP used to measure methylation level does not allow to understand what percent of DNA molecules are methylated and if this promoter is normally methylated in somatic cells. We think that a more direct method for DNA methylation analysis (eg bisulfite conversion-based assays) should be used at least in somatic tissues. Finally, the authors analyze methylation of other genes of the cluster but do not test the intergenic Dlk1-Meg3 DMR that is the imprinting control region of this domain.
4. There is some recent literature on human DIO3OS and cell proliferation, which may be useful to cite.

Minor points:

1. The authors use either the term "imprinting genes" or "imprinted genes". We think the latter is more appropriate.
2. Mitochondrial density (Fig. 1i) should be (Fig. 1I).

REVIEWER COMMENTS

Reviewer #1 (Remarks to the Author):

Manuscript: NCOMMS-21-04369 – “Imprinted LncRNA Dio3os preprograms intergenerational brown fat thermogenesis and obese resistance”. Chen Y-T et al.,

In this manuscript, Chen and colleagues investigated how maternal obesity (MO) predispose the offspring to obesity and metabolic dysfunctions. The authors found that MO inhibits the expression of a long noncoding Dio3 antisense RNA (Dio3os), resulting in enhanced expression of the Dio3 enzyme and reduction of intracellular T3 levels. In turns, the T3 deficiency impairs brown fat cell differentiation and fetal BAT development, favoring metabolic dysregulation and insulin resistance in MO female offspring.

Based on these new findings, the authors proposed that MO impairs female offspring BAT development and thermogenic function, contributing to insulin resistance and glucose intolerance.

The major claims are:

1. Maternal obesity (MO) renders female offspring metabolic dysfunction and cold sensitivity due to BAT dysregulation.
2. Impaired brown adipogenesis in MO female offspring has a fetal origin.
3. MO dysregulates maternally imprinted Dio3-Dio3os in fetal BAT, enhances Dio3 enzyme expression causing T3 deficiency.
4. Dio3os inactivation blocks brown adipogenic differentiation
5. Conversely, Dio3os activation in brown fat increases PRDM16 activity and body energy expenditure in female mice.
6. Dio3os activation in BAT prevents MO female offspring from diet-induced obesity
7. Dio3os hypermethylation mediates Dio3os suppression in obese dam oocytes and female offspring BAT.
8. The authors concluded that maternal obesity epigenetically regulates the expression of imprinted LncRNA Dio3os, which suppresses fetal BAT development and BAT functions.

Critique:

Overall, this is a well-written manuscript, presenting exciting new results and solid conclusions. However, there are some issues and questions that need to be addressed before the paper merit publication in Nature Communications (below):

Responses: Thanks for the reviewer's positive comments, and constructive suggestions to improve this manuscript.

Other Points:

1. The authors state that MO enhances the expression of deiodinase-3 (Dio3) and leads to intracellular T3 deficiency and impaired fetal BAT development. Due to the importance of

thyroid hormones in different tissues' development and function, it would be critical to examine whether MO also enhances *Dio3os* methylation, *Dio3* expression and causes T3 deficiency in other T3-regulated tissues such as the liver, skeletal muscle, heart, nerve.

Responses: Thanks for your suggestions. Following your suggestions, in the revision, we have analyzed *Dio3* and *Dio3os* expression in MO fetal and neonatal liver, limb skeletal muscle (SKM) and heart (**Supplementary Fig. 7b-e**). We found MO fetal SKM also had *Dio3os* lower expression and *Dio3* activation, aligned with lower T3 content in MO fetal SKM, while such changes were significantly blunted in neonates. Expression of *Dio3* and *Dio3os* was not altered in other organs of fetuses and neonates.

In previous studies, oxidative metabolism in heart, liver, brain and placenta was barely altered in thyroidectomized fetuses, while SKM and BAT were identified to be primary organs with oxidative malfunctions in fetal hypothyroidism [1-4], suggesting the growth and functions of fetal SKM and BAT highly rely on T3 availability and thyroid hormone (TH) signaling [1]. Because BAT and SKM are derived from the same progenitors (*Myf5+*) [5, 6], the *Dio3os* lower expression and reduced T3 content in both BAT and SKM of MO fetuses suggest that fetal BAT/SKM development is susceptible to MO.

Another interesting question is why higher *Dio3* expression and T3 deficiency were persistent in MO neonatal BAT, but not in SKM? Evolutionally, in order to prevent hypothermia at birth (neonates transitioning from the uterine $>37.5^{\circ}\text{C}$ to outside 22°C), BAT thermogenic activity is highly activated and peaked after birth [7]. Thermogenic activation requires T3 and TH signaling to activate mitochondrial biogenesis and uncoupling protein [8]. In comparison, after birth, such thermogenesis in SKM is marginal [9], which may attribute to the blunted *Dio3os* expression and no difference of T3 content in neonatal SKM due to MO.

Supplementary Fig. 7. b. mRNA expression of *Dio3* and *Dio3os* in liver, limb skeletal muscle (SKM) and heart of female fetuses ($n = 6$). **c.** T3 content in SKM in female fetuses ($n = 8$). **d.** mRNA expression of *Dio3* and *Dio3os* in liver, limb skeletal muscle (SKM) and heart of female neonates ($n = 6$). **e.** T3 content in SKM in female neonates ($n = 8$). Means \pm s.e.m., * $P < 0.05$; ** $P < 0.01$; *** $P < 0.001$.

2. Importantly, the authors have not addressed whether MO can also affect the browning of white adipose tissue (WAT) via a similar mechanism, such as T3 deficiency. Several reports indicating that WAT browning improves systemic insulin sensitivity and glucose metabolism have been published over the last decade. Thus, it would be essential to examine if MO affects *Dio3* expression, T3 levels, adipogenesis, and WAT browning induced by cold.

Response: Thanks for your suggestions. Following your suggestion, in the revision, we have measured UCP-1 and PGC-1 α proteins, browning level (UCP-1/ MitoTracker immunostaining), *Dio3*, *Dio3os* expression and T3 level in offspring inguinal and gonadal WAT under R.T. and following 4°C exposure (**Supplementary Fig. 4c-e** and **Supplementary Fig. 7f, g**). Data show that MO impaired inguinal-WAT browning in female offspring, but did not affect the expression of *Dio3* and *Dio3os*, and T3 content in WAT.

Supplementary Fig. 4. c. Immunoblotting measuring UCP-1, PGC-1a and VDAC proteins in female offspring inguinal-WAT at 22°C and following 4°C exposure. β -tubulin was used as a loading control ($n = 6$). **d,e** Immunostaining of UCP-1 (red) and mitochondria (green) in inguinal-WAT after 4°C exposure. Means \pm s.e.m., * $P < 0.05$; ** $P < 0.01$; *** $P < 0.001$.

Supplementary Fig. 7f, g. mRNA expression of *Dio3*, *Dio3os* (f) and T3 content (g) in inguinal- and gonadal-WAT of female offspring at weaning ($n = 6$). Means \pm s.e.m., * $P < 0.05$; ** $P < 0.01$; *** $P < 0.001$.

3. Also, as the T3 is essential for the nervous system function, the authors need to examine whether the MO-induced imprinting of LncRNA *Dio3os* affects the local production of T3 in other tissues besides BAT. Particularly the nerve system, which the normal development requires optimal levels of T4/T3. Moreover, abnormal sympathetic innervation and outflow in adipose tissue depots may impact BAT and WAT development, increasing the susceptibility to metabolic disorders.

Response: Thanks for your suggestions. In the revision, besides analyzing *Dio3os* and *Dio3* expression in other tissues (Supplementary Fig. 7b-g), we also have analyzed concentration of

epinephrine and norepinephrine in female offspring serum (**Supplementary Fig. 7h, i**), which are major neurohormones to activate BAT thermogenesis [10]. We also analyzed BAT and inguinal-WAT neural density through analyzing tyrosine hydroxylase (TH) protein (**Supplementary Fig. 7j**) [11, 12]. Besides, we did co-immunostaining of Dio3 protein with neural marker cholera toxin B (CTB) in BAT and inguinal-WAT to analyze Dio3 expression in BAT/WAT neurons. Because of lower immunostaining specificity of CTB in WAT, WAT-neurons were alternatively stained by TH as previously reported (**Supplementary Fig. 7k-n**) [13, 14]. Whole-mounting TH immunostaining was performed in inguinal-WAT for checking neural circuits, but not in BAT due to difficulties of available techniques for BAT lipid clearing (**Supplementary Fig. 7o**), which has been reported [12]. Results show that MO decreased neural density in inguinal-WAT of weaning offspring, but didn't affect concentrations of epinephrine and norepinephrine in serum and neural density in BAT of offspring. Dio3 immunostaining further showed that BAT/WAT neurons were weakly stained by Dio3 antibody but adipocytes were strongly stained. Taken together, these data suggest that MO impairs neural development in offspring WAT, while *Dio3* has a high expression primarily in brown adipocytes in MO female offspring BAT.

Supplementary Figure 7. h, i. Concentration of epinephrine and non-epinephrine in female offspring serum at P0 and 4-month-old ($n = 8$). **j.** Immunoblotting measuring tyrosine hydroxylase (TH) protein in neonatal offspring BAT and inguinal-WAT after weaning ($n = 6$). **k-n.** Co-immunostaining of Dio3 with cholera toxin B (CTB) or TH in offspring BAT (P0) and inguinal-WAT (P21). Intensity was quantified by image-J. Scale bar 400 μ m. **o.** Whole-mounting immunostaining of TH and Dio3 in offspring inguinal-WAT ($n = 4$). White arrow shows neural circuits in WAT. Scale bar 800 μ m. Means \pm s.e.m., * $P < 0.05$; ** $P < 0.01$.

4. Although the results in Figure 2 suggest BAT's importance in mediating the metabolic dysfunction of offspring MO female, it would be interesting to test whether AAV-mediated overexpression of Dio3os (AAV-Dio3os) can activate BAT even at thermoneutrality to improve systemic metabolism in MO offspring mice.

Response: Thanks for your suggestions. Following your suggestions, in the revision, we conducted an experiment to administrate MO female offspring with AAV-Dio3os in BAT under thermoneutrality at 30°C for 5 weeks (Fig. 7f-k and Supplementary Fig. 11c-j). MO offspring with AAV-Dio3os

increased BAT mitochondrial biogenesis, PGC-1a and UCP-1 proteins, substantially blunted body weight gain and adiposity. Meanwhile, MO offspring from AAV-Dio3os administration also improved glucose sensitivity, oxygen consumption and energy expenditure.

Fig. 7. h Normal and MO female neonatal offspring were administrated with PBS, AAV-SC or AAV-Dio3os in BAT (1.5×10^{11} vg per mouse) and further acclimated to

30°C for 5-wk ($n = 5$). BAT, inguinal- and gonadal WAT mass in offspring at 30°C. **i** Body mass in 5-wk at 30°C ($n = 5$). **j** Immunoblotting measuring PRDM16, UCP-1 and PGC-1a proteins in BAT at 30°C ($n = 5$). β -tubulin was as a loading control. **k, l** Immunostaining of UCP-1 and mito-tracker in BAT, and H&E staining of BAT, inguinal- and gonadal WAT (k). UCP-1, mitochondrial intensity and white adipocyte size were quantified by image-J (l). Scale bar 250 μ m (immunostaining) and 200 μ m (H&E). **m** Offspring glucose concentration after intraperitoneal glucose administration for glucose tolerance test at 30°C ($n = 5$). **n-p** O₂ consumption (n) and energy expenditure (o, p) of offspring at 30°C ($n = 5$). Metabolic data were regressed to body mass according to guidelines of ANCOVA NIDDK MCCP tool. Means \pm s.e.m., * $P < 0.05$; ** $P < 0.01$.

5. In Figure 3F and page 8 (line 164), the authors state that MO attenuated T3 levels in fetal and neonatal BAT. However, the information related to T3 levels in adipose tissue depots (BAT and WAT) from MO adult mice is missing. This info needs to be included.

Response: Thanks for your suggestions. In the revision, we have included T3 level in adult offspring BAT, inguinal- and gonadal-WAT (**Supplementary Fig. 12g and 7g**).

Supplementary Fig.12g and 7k. Contents of T3 in female offspring BAT at P21 and 4-month-old, and inguinal (subcutaneous) and visceral-WAT at P21. Means \pm s.e.m., * $P < 0.05$; ** $P < 0.01$.

6. Figure 4 and Figure S7 would be important to determine whether the binding of PGC1a to PRDM16 is also inhibited in MO adult female mice.

Response: Thanks for your suggestion. In the revision, we did a PRDM16 co-IP analysis in 4-month-old offspring BAT, and found that the reduced binding affinity of PRDM16 with PGC-1a were persistent in adults (**Supplementary Fig. 8g**).

Supplementary Fig. 8g. PRDM16 immunoprecipitation in measuring PGC-1a and CtBP1 binding proteins in isolated brown preadipocytes in 4-month-old offspring BAT ($n = 5$). PRDM16, PGC-1a and CtBP1 were detected by immunoblotting. IgG was used as a negative control in immunoprecipitation. Means \pm s.e.m., ** $P < 0.01$; *** $P < 0.001$.

7. In Figure 6, the UCP1 expression in BAT from MO female mice was not reduced compared to control. Moreover, AAV-Dio3os administration did not seem to affect much UCP1 expression in BAT from MO mice. The quantification of UCP1 and other thermogenic protein levels in control, MO, PBS-treated versus AAV-Dio3os-treated mice should be conducted.

Response: Thanks for your suggestions. We increased sample number and re-analyzed UCP-1 expression, and found AAV-Dio3 administration significantly rescued UCP-1 expression in MO offspring. The quantified data were also included in the revised manuscript (**Supplementary Fig. 9e**).

Supplementary Fig. 9e. Quantified adipogenic and thermogenic proteins in offspring BAT at 22°C ($n = 5$). Means \pm s.e.m., * $P < 0.05$; ** $P < 0.01$; *** $P < 0.001$.

8. As the higher methylation of Dio3os promoter and reduced Dio3os expression was also observed in adult BAT (Figure 7d), the authors should examine thyroid hormone levels and THR signaling in BAT from MO adult mice.

Response: Thanks for your suggestions. In the revised manuscript, we also included T3 level and THR expression in MO offspring BAT at weaning and 4-month-old (**Supplementary Fig. 12 g, h**).

Supplementary Fig. 12. g. T3 concentration in BAT of female offspring at weaning (P21) and 4 months of age ($n = 6$). **h.** mRNA expression of thyroid hormone receptors in female offspring BAT at weaning and 4 months of age ($n = 6$). Means \pm s.e.m., * $P < 0.05$; ** $P < 0.01$; *** $P < 0.001$.

9. The authors state that recovering Dio3os expression in MO female offspring increases BAT thermogenesis, preventing offspring from western diet-induced obesity and metabolic dysfunctions. However, they did not investigate the effect of AAV-Dio3os-OE on BAT expansion in prolonged cold exposure (i.e., 4 days at 4°C, as shown in Figure 2).

Response: Thanks for your suggestions. Following your suggestions, in the revision, we included BAT thermogenesis, glucose sensitivity and cold resistance in MO female offspring with AAV-Dio3os administration at 4°C for 4 days (**Fig. 7a-e and Supplementary Fig. 11 a, b**). Results showed that AAV-Dio3 administration substantially recovered BAT adipogenesis and thermogenic activation in MO offspring, improving glucose sensitivity and cold resistance.

Figure 7. a After 4°C exposure for 4 days, BAT mass in 8-week-old normal and MO female offspring administrated with PBS, AAV8-scramble-CMV or AAV8-Dio3os-CMV viral particles (1.5×10^{11} vg per

mouse, $n = 6$). **b** Immunoblotting measuring PRDM16, PGC-1a and UCP-1 proteins in BAT after 4°C exposure ($n = 6$). β -Tubulin was as a loading control. ($n = 6$). **c** Heatmap displaying mRNA expression of brown thermogenic and mitochondrial biogenic genes in BAT after 4°C exposure ($n = 6$). mRNA expression was normalized to 18S rRNA. **d** mtDNA copy number in BAT after 4°C exposure ($n = 6$). **e** Blood glucose in offspring after intraperitoneal glucose injection for measuring glucose tolerance at 4°C ($n = 6$). **f, g** Interscapular surface and rectal body temperature in offspring at 4°C ($n = 6$). Means \pm s.e.m., * $P < 0.05$; ** $P < 0.01$; *** $P < 0.001$; **** $P < 0.0001$.

Reviewer #2 (Remarks to the Author):

The prevalence of obesity is increasing, even among women of childbearing age. In addition to the short-term complications of obesity for both mother and child, emerging evidence suggests that maternal obesity has long-term detrimental consequences for offspring health. Though multiple genome-wide association studies have been carried out in the past decades to try to identify the genetic basis of these disorders, the factors responsible for these diseases and their heritability are still obscure. Furthermore, nongenetic components, specifically epigenetic factors, has recently been recognized, and may also contribute to offspring traits acquired. In this manuscript, the author reports maternal obesogenic diet before mating and during gestation renders female offspring metabolic dysfunction and cold intolerance, and they showed lower BAT mass and downregulated genes involved in brown adipogenesis in MO female offspring, and they were originated from fetal stage. Mechanistically, enhanced expression of Dio3 inactive T3 to T2, which leads to intracellular T3 deficiency and suppression of BAT development. To show the regulatory role of Dio3os for Dio3 expression, the authors provide the evidence: in vitro knockdown of Dio3os activated Dio3 expression, reduced intracellular T3 availability and expression of Thra/Thrb, and blocked brown adipogenic differentiation; in vivo overexpression of Dio3os reduced D3 content, increased BAT mass, and activated BAT thermogenesis, which consequently prevented MO female from western diet induced obesity. Furthermore, they show high methylation of Dio3os promoter contributes to the reduced Dio3os expression in MO offspring BAT. The authors propose that methylation changes in fetal and adult BAT were originated from oocytes, and conclude that methylation in Dio3os promoter may serve as a novel imprinted LncRNA in mediating intergenerational obesity.

This study is performed on the MO induced fetal BAT phenotypic analysis, and presents the evidence to support Dio3 dysregulation mediates maternal obesity induced fetal BAT development defects. Dio3 function has been well recognized. Some findings, for example, Dio3os suppresses Dio3 expression, and its expression is controlled by methylation in promoter region that behaves like a maternal imprinted gene, are interesting. However, the molecular mechanism underlying this process has not been clearly presented here, which is important for people to understand the intergenerational effects of BAT in obesity.

Major concerns:

1. In the study, they showed that brown progenitors and preadipocytes were profoundly reduced in MO fetal BAT, and isolated SVFs in MO fetal BAT had lowered capacity to differentiate into mature brown adipocytes in vitro, consequently decreasing BAT mass. Based on these results, metabolic dysfunction in female offspring is mainly due to impaired brown adipogenesis. Considering that disruption of brown fat thermogenesis is caused by deficiency of mature brown

adipocytes, the current title of manuscript may be changed to “...brown fat development or differentiation”?

Response: Thanks for your comment. In the revision, title has been changed to “brown fat development”.

2. Prdm16 controls a bidirectional cell fate switch between skeletal myoblasts and brown fat cells. Loss of Prdm16 from brown fat precursors results in a loss of brown fat characteristics and promotes muscle differentiation (Nature 2008. 21; 454(7207):961-7). The authors showed that the expression of PRDM16 was suppressed in MO offspring, so for UCP-1 and PGC1-1a protein contents and expression of thermogenic genes. It is better to detect whether the expression of muscle-specific genes are upregulated when PRDM16 is downregulated.

Response: Thanks for your comment. Following your suggestion, in the revision, we have analyzed muscle specific genes in fetal and neonatal BAT, including Pax7, Myod, Myog, MCK and Myf6 (Supplementary Fig. 5h).

Supplementary Fig. 5h. mRNA expression of myogenic regulatory genes in offspring BAT at E18.5 and P0 ($n = 6$). Means \pm s.e.m., * $P < 0.05$; ** $P < 0.01$; *** $P < 0.001$.

3. In Fig. 1m, expression of genes involved in lipid uptake was decreased in MO offspring BAT; however, in Fig 1l, transmission electron microscopy image showed that lipid contents seem to be accumulated in MO offspring. how to explain that?

Response: Thanks for your comment. Brown adipocytes are a special type of fat cells, which not only have a high capacity to uptake fatty acids in blood circulation, are also highly dependent on enriched mitochondria to actively oxidize fatty acids to generate uncoupling heat. MO offspring BAT showed a substantial reduction of mitochondrial gene expression (PGC-1a etc.) and mitochondrial number (Figure 1j, l, m and Supplementary Fig. 3g), which impairs fatty acid oxidation and contributes to excessive lipid accumulations in brown adipocytes. In the revised manuscript, we included the information to make the results clearer.

Fig. 1k. PGC-1a protein in BAT in female offspring from chow or HFD feeding.

Supplementary Figure 3g. Quantified mitochondrial number in female offspring BAT. Means \pm s.e.m., * $P < 0.05$; ** $P < 0.01$; *** $P < 0.001$

4. The authors declare thermogenesis, lipid uptake and oxidization were decreased in MO offspring BAT, contributing to lipid accumulates in WAT and liver (Fig 1n and Fig. 6f). Lipid drop deposited in WAT and liver was due to BAT dysfunction or metabolic defects occur in WAT and liver? To clear this, it is needed to characterize the metabolic traits of WAT and liver.

Response: Thanks for your suggestion. In the revision, we measured insulin signaling, mitochondrial biomass and respiration in inguinal WAT and liver of MO offspring with and without AAV-Dio3os administration in BAT (**Supplementary Fig. 10g-k**). Results showed that AAV-Dio3os administration in BAT significantly recovered the impaired insulin sensitivity, mitochondrial biogenesis and respiration in MO offspring inguinal-WAT and liver, suggesting BAT thermogenic activation in offspring plays critical roles in attributing to WAT and hepatic metabolic health.

Supplementary Fig. 10 g. Immunoblotting measuring IRS-1 (T612) and AKT (S473) phosphorylation in offspring inguinal WAT and liver stimulated by intraperitoneal insulin administration (1 U/kg) after 30 min ($n = 5$). β-tubulin was as a loading control. **h.** Immunoblotting measuring VDAC protein in inguinal-WAT and liver in offspring with scramble or AAV-Dio3os administration in BAT ($n = 6$). **i-k.** Fluorescent assays measuring extracellular oxygen consumption rate (OCR) in inguinal WAT and liver in offspring *ex vivo* ($n = 5$). Tissues were freshly prepared and cultured in DMEM at 37°C, and OCR was normalized to tissue mass. Means ± s.e.m., * $P < 0.05$; ** $P < 0.01$; *** $P < 0.001$

5. Based on the findings in this report, T3 deficiency was occurred in BAT, not in serum. In supplementary Figure 6c-h, the results showed T4, T3 and TSH concentrations in serum were not significantly altered in neither female fetuses nor neonates. However, this is likely due to the great variance among individuals. Increasing sample size and re-analysis of the data may be needed to clarify this issue.

Response: Thanks for your suggestion. In the revision, we increased the sample size to analyze thyroid hormone concentration in both fetal and neonatal BAT and serum (**Fig. 3d and Supplementary Fig. 5i**), and showed the T3 deficiency occurred in MO offspring BAT.

Fig. 3d T4 and T3 contents in fetal and neonatal BAT ($n = 17-20$).

Supplementary Fig. 5i Concentration of T3, T4 and TSH in fetal and neonatal serum ($n = 11$). Means \pm s.e.m., * $P < 0.05$; ** $P < 0.01$; *** $P < 0.001$

6. Dio3os overexpression in interscapular BAT activates BAT adipogenesis and thermogenesis, and prevents female offspring from diet induced obesity. Is it possible that AAV-Dio3os vector leaks into circulation and functions in tissues response to thyroid hormone, like WAT and liver?

Response: Thanks for your suggestion. In the revision, we analyzed the *Dio3os* expression in inguinal-WAT, liver and skeletal muscle in mice administrated with AAV-Dio3os in BAT (**Supplementary Fig 10a**), and showed no alteration in WAT, liver and skeletal muscle.

Supplementary Fig. 10a. *Dio3os* and *Dio3* mRNA expression in liver, tibialis anterior muscle (TA) and inguinal WAT in mice with scramble or AAV-Dio3os administration in BAT ($n = 5$). Means \pm s.e.m.

7. The association between Dio3os inactivation and Dio3 overexpression is extensively examined in this report; however, much of the data is correlative. Dio3 and Dio3os locate in the DLK1-Dio3 imprinting locus. Dio3os, also named Dio3 opposite strand, locates in 1.6 kb upstream of Dio3, expresses from the maternal allele (Development, 2007, 134, 417-426). Dio3, a maternal imprinting gene, expresses from paternal allele. Here, the authors detected the expression of other imprinting genes (both paternally expressed) in the locus, like *Dlk1* and *Rtl1*. How about the expression of other maternal imprinting genes located in this locus, such as *Gtl2* and some microRNA? It is also very essential to figure out which allele (maternal or paternal) contributes to the Dio3 expression? Importantly, it is still confusing that how suppression of Dio3os activates the expression of Dio3? By what mechanisms? It is a great shame that these questions are still open in this study.

Response: Thanks for your comment. Per your suggestion, we analyzed *Gtl2* and major microRNA expressions in the DLK1-Dio3 locus (**Supplementary Fig. 7a**), and *Gtl2* and most of

miRNA expressions were barely altered in MO offspring BAT; *per your comment*, we also included information and references that *Dio3* is a paternally expressed gene in line 184-185 (Results section) and line 377-380 (Discussion section) in the revised manuscript;

Supplementary Fig. 7a. mRNA expression of *Glt2* and miRNAs in BAT of female fetuses and neonates ($n = 5-6$). miRNA expression was normalized to U6. Means \pm s.e.m., * $P < 0.05$.

per your comment, to solidify the causal effect, we overexpressed either *Dio3os*, or *Dio3* or both in mouse embryonic fibroblasts (MEFs), which was followed by brown adipogenic induction (**Supplementary Figure 9k-o**). Results show that *Dio3* overexpression abrogated *Dio3os*-induced brown adipogenic activation, such as brown adipocyte formation, thermogenic gene activation and mitochondrial respiration, showing *Dio3* suppression is mainly responsible for *Dio3os* in promoting brown adipogenesis.

Supplementary Fig. 9 Mouse embryonic fibroblasts (MEFs) were transfected with scramble, or AAV8-*Dio3os*-CMV viral particles, or *Dio3* ORF plasmid, or both following brown adipocyte induction for 5 days ($n = 4$). **k.** Oil-Red O staining in differentiated brown adipocytes ($n = 4$). Lipids were quantified by microplate reader at 492 nm ($n = 4$). **l.** Immunoblotting measurements of PRDM16, PGC-1a and UCP-1 proteins in

differentiated brown adipocytes ($n = 4$). β -tubulin was used as a loading control. **m.** Immunostaining of UCP-1 and mitochondria in differentiated brown adipocytes. Intensity was quantified by image-J. **n, o.** Extracellular oxygen consumption of differentiated brown adipocytes measured by fluorescent assays. Means \pm s.e.m., * $P < 0.05$; ** $P < 0.01$; *** $P < 0.001$; **** $P < 0.0001$.

Regarding to *Dio3os* mechanistic regulations, miR-122 has been reported to activate gene expressions (miRNAs) in the *Dlk1-Dio3* locus [15], and cause excessive lipid accumulations in WAT and liver [16], linking to adult obesity and glucose tolerance. Furthermore, lncRNA *Dio3os* was shown to directly interact with miR-122 and inhibited its activity [17]. In MO female offspring BAT, we also observed a high activation of miR-122-3/5p and *Dio3*, and reduced expression of *Dio3os* in BAT from MO fetuses and offspring. Considering a versatility of miRNAs in regulating gene expression, such as sponge roles in transcriptional control, whether miR-122 critically mediates *Dio3* activation by *Dio3os*, and more mechanistic studies will be conducted in future studies. Related discussions have been added in the manuscript.

Fig. a miR-122 transcription in offspring BAT from embryonic 18.5 days (E18.5) to 4-month-old. **b.** Expression of miR-122-3p and miR-122-5p in BAT of offspring born from normal and obese dam. Expression was normalized to U6. Means \pm s.e.m., * $P < 0.05$; ** $P < 0.01$; *** $P < 0.001$; **** $P < 0.0001$.

8. The authors conclude that methylation changes in *Dio3os* promoter can persist to early embryogenesis and adult tissue, consequently contributing to the metabolic phenotypes in the offspring. Hence, it should not be a tissue specific imprinting. All tissues from MO offspring should be hypermethylated in *Dio3os* promoter, and the expression of *Dio3* and *Dio3os* may be altered accordingly. In the present study, the authors found that MO male offspring are less susceptible to body weight gain and insulin resistance compared to female. They think this gender dimorphism is resulted from the differential effects of sex hormone. Since *Dio3os* is maternal imprinting gene as the authors stated in the manuscript, *Dio3os* promoter in the tissues of male offspring may also acquire the similar methylation pattern, and the expression of *Dio3* and *Dio3os* might be changed. So detection of DNA methylation and gene expression in other female tissues and male BAT would further strengthen the concept of maternal imprinting.

Responses: Thanks for your insightful comments and constructive suggestions. In the revision, per your suggestion, we analyzed *Dio3* and *Dio3os* expressions in female offspring liver, heart, skeletal muscle and inguinal and gonadal-WAT, and male offspring BAT (**Supplementary Fig. 7b-g and p**), and results show that expression of *Dio3* and *Dio3os* was altered in the limb skeletal muscle of MO female fetuses, while was barely altered in other tissues/organs and male BAT. Based on current knowledge, DNA methylation of imprinted genes in germlines is known to escape from epigenetic resetting during embryonic and fetal development, but also changes during development, resulting in tissue- and sex-dependent changes in imprinting, which consequently drive the distinct morphology/physiology/diseases and metabolism among tissues

and sexes [18-20]. For the sexual dimorphism, in addition to sexual hormones, Day and Bondurianky's sexual antagonism theory for imprinting gene expression (<https://www.nature.com/articles/hdy201429>) shows that female offspring overall favor expressing maternal imprinted genes (matrigenic expression), while male offspring favor expressing paternal imprinted genes (patrigenic expression) [21]. Fetal programming of maternal obesity or diabetes was also shown to be mostly transmitted through maternal line across multiple generations in previous studies [22]. In the revised manuscript, we rephrased the statement, and suggested higher methylation of *Dio3os* promoter likely/potentially contributes to *Dio3os* inactivation in BAT (line 84; line 321; line 397). Besides, we also included the Day and Bondurianky's antagonism imprinted theory for explaining the sexual dimorphism of imprinted genes.

Supplementary Fig. 7. b. mRNA expression of *Dio3* and *Dio3os* in liver, limb skeletal muscle (SKM) and heart of female fetuses ($n = 6$). **c.** T3 content in SKM in female fetuses ($n = 8$). **d.** mRNA expression of *Dio3* and *Dio3os* in liver, limb skeletal muscle (SKM) and heart of female neonates ($n = 6$). **e.** T3 content in SKM in female neonates ($n = 8$). **p.** mRNA expression of *Dio3* and *Dio3os* in male offspring BAT at 18.5 and P0 ($n = 6$). Means \pm s.e.m., * $P < 0.05$; ** $P < 0.001$.

Minor points

1. Some errors in figures: line 108 should be “Fig. 1h and supplemental Fig. 3e,f”; line 110 should be “Fig. 1l and supplemental Fig. 3g”;

Response: Thanks for your suggestion. The errors have been corrected in the revised manuscript.

2. Fig. 2b, BAT mass gain should be presented as relative value.

Responses: Thanks for your suggestion. BAT mass gain has been changed to relative value in the revised manuscript.

3. Fig. 2e, the figure was not correctly labeled, 22C?

Response: Thanks for your suggestion. It has been corrected to 22°C in the revised manuscript.

4. Mitochondrial density in Fig. 1l should be quantified. Similar issues can be found in many figures in this paper, such as Fig. 3I and other figures (HE and immunofluorescence pictures).

Response: Thanks for your suggestion. The mitochondrial density (Supplementary Fig. 3g) and fluorescent intensity in immunostaining and adipocyte size in H.E. images have been quantified and presented in the revised manuscript (Supplementary Fig. 5e, 10b-e, 11i and j).

5. Oocyte numbers in Fig. 7 should be included in the Figure legends.

Response: Thanks for your suggestion, oocyte number has been included in the figure legend.

Reviewer #3 (Remarks to the Author):

This study investigates the role of the lncRNA Dio3os in the control of brown fat differentiation and thermogenesis and development of obesity in the mouse. In brief, the authors demonstrated that the female offspring of mice fed with high-fat diet during pregnancy developed obesity and metabolic disorder, which were associated with less BAT and more WAT deposition. Consistent with this finding, this progeny accumulated less BAT and had reduced thermogenic capacity when exposed to cold temperatures. To look for the origin of this metabolic alteration, the authors also found that BAT reduction began at fetal stages and was associated with impaired differentiation capacity of brown fat progenitor cells. The cause of this abnormality was identified in a lower level of the thyroid hormone T3 that in turn was associated with higher expression of the deiodinase Dio3 gene. Because Dio3 is close to the lncRNA gene Dio3os, the authors investigated its expression and found that it was down-regulated in brown fat. By employing a knockdown model in mouse embryonic fibroblasts, they demonstrated that Dio3os negatively regulated Dio3 expression and inhibited in vitro brown adipocyte differentiation that could be rescued by T3 supplementation. Inhibition of Dio3os in a further ex-vivo model obtained with stromal vascular cells isolated from fetal BAT led to similar conclusions. They then forced Dio3os expression by injecting a transgene vehiculated by an adenovirus-based vector into intra-scapular BAT. They found that such treatment decreased Dio3 and increased T3 and brown adipogenesis also reducing the gain of body weight in the mice fed with a normal diet and preventing the development of obesity and metabolic dysfunction in the offspring of obese female fed with a high-fat diet. Finally, they found that Dio3os suppression in fetal BAT of obese dams was associated with the gain of methylation of the Dio3os promoter. Higher Dio3os promoter methylation was found also in adult BAT of the obese mice offsprings and in the oocytes of the obese dams, leading the authors to conclude that maternal methylation persisted in their offspring.

This study provides several new and useful information. However, we think there are some points that can be improved or clarified.

1. I am confused with the different types of high-fat diets used in this study: HFD, western diet, obesogenic diet. Please, clarify the need to use different diets.

Response: Thanks for your suggestion. In the revised manuscript, we throughout used HFD and included % (calorie from fat) when it is necessary.

2. The issue of gender dimorphism in the development of obesity is interesting but not properly investigated. What is the molecular mechanism causing this gender difference? Is Dio3os downregulated and methylated and Dio3 upregulated also in the male offspring of obese dams? Or is the difference more downstream?

Response: Thanks for your suggestion. Although the molecular mechanisms of this gender difference remain obscure, from the current understanding, in addition to differences in sexual

hormones between male and females, Day and Bondurianky's sexual antagonism theory (<https://www.nature.com/articles/hdy201429>) shows that female offspring favor expressing maternal imprinted genes (matrigenic expression), while male offspring favor expressing paternal imprinted genes (patrigenic expression) [21]. Because of *Dio3os* dominantly expresses from maternal allele, it may contribute to female offspring BAT susceptible to *Dio3os* inactivation. Besides, fetal programming of maternal obesity or diabetes was also shown to be mostly transmitted through maternal line across multiple generations in previous studies [22]. The information has been also included in the revised manuscript. *Per your suggestion*, we also analyzed *Dio3os* and *Dio3* expression in male fetal and neonatal BAT (**Supplementary Fig. 7q**).

Supplementary Fig.7q mRNA expression of *Dio3os* and *Dio3* in male offspring BAT at E18.5 and P0 ($n = 6$).

3. I could not find any study demonstrating the imprinting of *Dio3os* and even imprinting of *Dio3* is known to be tissue-specific. Because the authors do not investigate the allele-specific expression of these genes, I think it is improper to refer to their imprinted expression, as they did in several figures. Also, *Dio3os* promoter methylation could be the consequence rather than the cause of *Dio3os* repression and the MeDIP used to measure methylation level does not allow to understand what percent of DNA molecules are methylated and if this promoter is normally methylated in somatic cells. We think that a more direct method for DNA methylation analysis (eg bisulfite conversion-based assays) should be used at least in somatic tissues. Finally, the authors analyze methylation of other genes of the cluster but do not test the intergenic *Dlk1-Meg3* DMR that is the imprinting control region of this domain.

Responses: Thanks for your suggestions. *Per your comment*, we rephrased the “imprinted expression” to “gene expression” in the revised manuscript. *Per your comment*, we also applied the bisulfite pyrosequencing to quantify the CpG methylation levels in the *Dio3os* promoter region and IG-DMR in offspring BAT at E18, P0, P21 and 4 months old (**Fig. 8 d, e and Supplementary Fig. 12c**). Bisulfite pyrosequencing analysis showed a significant increase of CpG methylation in MO offspring BAT, which was persistent in offspring mice at different ages. The CpG methylation in IG-DMR was barely altered by MO, which was consistent with no significant change in expression of other imprinted genes closer to the IG-DMR, such as *Dlk1*, *Gtl2* and *Rtl1*.

Fig. 8. d-f Diagram shows bisulfite pyrosequencing assayed the genomic location and percentage of CpG methylation in the *Dio3os* promoter (d). CpG methylation percentage in the *Dio3os* promoter in offspring BAT at E18.5 ($n = 8$), birth ($n = 10$), weaning ($n = 6$) and 4-month-old ($n = 8$) (e). **f** Dynamics of average CpG methylation percentages in the *Dio3os* promoter in offspring BAT measured by bisulfite pyrosequencing. **Supplementary Fig. 12c.** Bisulfite pyrosequencing assayed CpG methylation in IG-DMR in offspring BAT from E18.5 to 4-month-old. Means \pm s.e.m., * $P < 0.05$; ** $P < 0.01$; *** $P < 0.001$; **** $P < 0.0001$.

4. There is some recent literature on human DIO3OS and cell proliferation, which may be useful to cite.

Response: Thanks for your comment. Following your suggestion, recent literatures on human *Dio3os* and cell proliferation have been cited and included in the revised manuscript line 385-388.

Minor points:

1. The authors use either the term “imprinting genes” or “imprinted genes”. We think the latter is more appropriate.

Response: Thanks for your suggestion, “imprinting genes” has been corrected to “imprinted genes” throughout the manuscript.

2. Mitochondrial density (Fig. 1i) should be (Fig. 11).

Response: Thanks for your suggestion, Fig.1i has been corrected to Fig. 11.

References

1. Forhead AJ, Fowden AL. Thyroid hormones in fetal growth and prepartum maturation. *Journal of Endocrinology*. 2014;221(3):R87-R103. doi: 10.1530/JOE-14-0025.
2. Polk DH, Callegari CC, Newnham J, Padbury JF, Reviczky A, Fisher DA, et al. Effect of fetal thyroidectomy on newborn thermogenesis in lambs. *Pediatric Research*. 1987;21:453-7.
3. Bhakthavathsalan A, Mann LI, Ayromlooi J, Kunzel W, Liu M. The effects of fetal thyroidectomy in the ovine fetus. *American Journal of Obstetrics and Gynecology*. 1977;127(3):278-84. doi: [https://doi.org/10.1016/0002-9378\(77\)90469-0](https://doi.org/10.1016/0002-9378(77)90469-0).
4. Herpin P, Berthin D, Duchamp C, Dauncey MJ, Dividich JL. Effect of thyroid status in the perinatal period on oxidative capacities and mitochondrial respiration in porcine liver and skeletal muscle. *Reproduction, Fertility, and Development*. 1996;8:147-55.
5. Seale P, Bjork B, Yang W, Kajimura S, Chin S, Kuang S, et al. PRDM16 controls a brown fat/skeletal muscle switch. *Nature*. 2008;454(7207):961-7. Epub 2008/08/23. doi: 10.1038/nature07182. PubMed PMID: 18719582; PubMed Central PMCID: PMC2583329.
6. Farmer SR. Brown Fat and Skeletal Muscle: Unlikely Cousins? *Cell*. 2008;134(5):726-7. doi: <https://doi.org/10.1016/j.cell.2008.08.018>.
7. Cannon B, Nedergaard JAN. Brown Adipose Tissue: Function and Physiological Significance. *Physiological Reviews*. 2004;84(1):277-359. doi: 10.1152/physrev.00015.2003.
8. Bianco AC, McAninch EA. The role of thyroid hormone and brown adipose tissue in energy homeostasis. *Lancet Diabetes Endocrinol*. 2013;1(3):250-8. Epub 2013/10/18. doi: 10.1016/S2213-8587(13)70069-X. PubMed PMID: 24622373.
9. Zhang D, Wang X, Li Y, Zhao L, Lu M, Yao X, et al. Thyroid hormone regulates muscle fiber type conversion via miR-133a1. *J Cell Biol*. 2014;207(6):753-66. doi: 10.1083/jcb.201406068.
10. Schulz TJ, Tseng YH. Systemic control of brown fat thermogenesis: integration of peripheral and central signals. *Ann N Y Acad Sci*. 2013;1302(1):35-41. Epub 2013/10/12. doi: 10.1111/nyas.12277. PubMed PMID: 24111913; PubMed Central PMCID: PMC3805713.
11. Murano I, Barbatelli G, Giordano A, Cinti S. Noradrenergic parenchymal nerve fiber branching after cold acclimatisation correlates with brown adipocyte density in mouse adipose organ. *J Anat*. 2009;214(1):171-8. Epub 2008/11/12. doi: 10.1111/j.1469-7580.2008.01001.x. PubMed PMID: 19018882.
12. Jiang H, Ding X, Cao Y, Wang H, Zeng W. Dense Intra-adipose Sympathetic Arborizations Are Essential for Cold-Induced Beiging of Mouse White Adipose Tissue. *Cell Metab*. 2017;26(4):686-92.e3. Epub 2017/09/19. doi: 10.1016/j.cmet.2017.08.016. PubMed PMID: 28918935.
13. Morrison SF, Madden CJ. Central nervous system regulation of brown adipose tissue. *Comprehensive Physiology*. 2014;4(4):1677-713. doi: 10.1002/cphy.c140013. PubMed PMID: 25428857.
14. Jeong JH, Lee DK, Blouet C, Ruiz HH, Buettner C, Chua S, et al. Cholinergic neurons in the dorsomedial hypothalamus regulate mouse brown adipose tissue metabolism. *Mol Metab*. 2015;4(6):483-92. doi: <https://doi.org/10.1016/j.molmet.2015.03.006>.

15. Valdmanis PN, Kim HK, Chu K, Zhang F, Xu J, Munding EM, et al. miR-122 removal in the liver activates imprinted microRNAs and enables more effective microRNA-mediated gene repression. *Nature Communications*. 2018;9(1):5321. doi: 10.1038/s41467-018-07786-7.
16. Esau C, Davis S, Murray SF, Yu XX, Pandey SK, Pear M, et al. miR-122 regulation of lipid metabolism revealed by in vivo antisense targeting. *Cell Metabolism*. 2006;3(2):87-98. doi: <https://doi.org/10.1016/j.cmet.2006.01.005>.
17. Cui K, Jin S, Du Y, Yu J, Feng H, Fan Q, et al. Long noncoding RNA DIO3OS interacts with miR-122 to promote proliferation and invasion of pancreatic cancer cells through upregulating ALDOA. *Cancer Cell Int*. 2019;19:202. Epub 2019/08/07. doi: 10.1186/s12935-019-0922-y. PubMed PMID: 31384177; PubMed Central PMCID: PMC6668142.
18. Weinstein LS. The role of tissue-specific imprinting as a source of phenotypic heterogeneity in human disease. *Biological Psychiatry*. 2001;50(12):927-31. doi: [https://doi.org/10.1016/S0006-3223\(01\)01295-1](https://doi.org/10.1016/S0006-3223(01)01295-1).
19. Lau JC, Hanel ML, Wevrick R. Tissue-specific and imprinted epigenetic modifications of the human NDN gene. *Nucleic Acids Res*. 2004;32(11):3376-82. Epub 2004/07/13. doi: 10.1093/nar/gkh671. PubMed PMID: 15247330; PubMed Central PMCID: PMC443546.
20. Biliya S, Bulla LA, Jr. Genomic imprinting: the influence of differential methylation in the two sexes. *Exp Biol Med (Maywood)*. 2010;235(2):139-47. Epub 2010/04/21. doi: 10.1258/ebm.2009.009251. PubMed PMID: 20404028.
21. Patten MM, Ross L, Curley JP, Queller DC, Bonduriansky R, Wolf JB. The evolution of genomic imprinting: theories, predictions and empirical tests. *Heredity*. 2014;113(2):119-28. doi: 10.1038/hdy.2014.29.
22. Hanafi MY, Abdelkhalek TM, Saad MI, Saleh MM, Haiba MM, Kamel MA. Diabetes-induced perturbations are subject to intergenerational transmission through maternal line. *J Physiol Biochem*. 2016;72(2):315-26. Epub 2016/04/03. doi: 10.1007/s13105-016-0483-7. PubMed PMID: 27038466.

Reviewers' Comments:

Reviewer #1:

Remarks to the Author:

The authors have carefully addressed the issues and questions raised by this review. The new results and revisions made by the authors have significantly improved the quality of this manuscript. I am satisfied with the rewrites, new data, and comments to my concerns.

Reviewer #2:

Remarks to the Author:

I find the current version of the manuscript improved and most of my criticisms addressed. I support its publication in Nature Communications but some descriptions in this manuscript need to be modified.

1. Line72, Dio3os is a paternally imprinted but not a 'maternally imprinted LncRNA'.
2. Line90, one of the 'after weaning' should be deleted.
3. Line147, the description of 'maternally imprinted Dio3-Dio3os' in here is inappropriate, because Dio3os is paternally imprinted LncRNA. Besides, Dio3os's expression and function are not detected in this section.
4. Line216, 'Dio3os-Dio3 imprinted locus' in here is misuse. Imprinted locus usually refers to multiple genes under coordinated epigenetic control. Is the expression of Dio3os and Dio3 controlled by the same genomic locus?

Reviewer #3:

Remarks to the Author:

In general, I am satisfied with the changes introduced by the authors into the revised manuscript. My last request is to use the terms "mat" and "pat" rather than the gender symbols to indicate the maternal and paternal alleles of the imprinted genes in the Figures (including Supplemental Figs).

REVIEWERS' COMMENTS

Reviewer #1 (Remarks to the Author):

The authors have carefully addressed the issues and questions raised by this review. The new results and revisions made by the authors have significantly improved the quality of this manuscript. I am satisfied with the rewrites, new data, and comments to my concerns.

Reply: thanks for the reviewer's constructive and insightful suggestions to improve the manuscript.

Reviewer #2 (Remarks to the Author):

I find the current version of the manuscript improved and most of my criticisms addressed. I support its publication in Nature Communications but some descriptions in this manuscript need to be modified.

1. Line72, Dio3os is a paternally imprinted but not a 'maternally imprinted LncRNA'.

Reply: following your suggestion, the sentence has been corrected.

2. Line90, one of the 'after weaning' should be deleted.

Reply: following your suggestion, one of "after weaning" has been removed.

3. Line147, the description of 'maternally imprinted Dio3-Dio3os' in here is inappropriate, because Dio3os is paternally imprinted LncRNA. Besides, Dio3os's expression and function are not detected in this section.

Reply: thanks for your suggestion, we corrected 'maternally imprinted Dio3-Dio3os' to "T3" in subtitle.

4. Line216, 'Dio3os-Dio3 imprinted locus' in here is misuse. Imprinted locus usually refers to multiple genes under coordinated epigenetic control. Is the expression of Dio3os and Dio3 controlled by the same genomic locus?

Reply: thanks for your suggestion, we revised and removed "Dio3os-Dio3 imprinted locus" in the sentence.

Reviewer #3 (Remarks to the Author):

In general, I am satisfied with the changes introduced by the authors into the revised manuscript. My last request is to use the terms "mat" and "pat" rather than the gender symbols to indicate the maternal and paternal alleles of the imprinted genes in the Figures (including Supplemental Figs).

Reply: Thanks for your suggestion, we have replaced terms to "mat" and "pat" in figures.